# 3-Vinylazetidin-2-Ones: Synthesis, Antiproliferative and Tubulin Destabilizing Activity in MCF-7 and MDA-MB-231 Breast Cancer Cells

**DOI:** 10.3390/ph12020056

**Published:** 2019-04-11

**Authors:** Shu Wang, Azizah M. Malebari, Thomas F. Greene, Niamh M. O’Boyle, Darren Fayne, Seema M. Nathwani, Brendan Twamley, Thomas McCabe, Niall O. Keely, Daniela M. Zisterer, Mary J. Meegan

**Affiliations:** 1School of Pharmacy and Pharmaceutical Sciences, Trinity College Dublin, Trinity Biomedical Sciences Institute, 152-160 Pearse Street, Dublin 2 DO2R590, Ireland; wangsh@tcd.ie (S.W.); tgreene@tcd.ie (T.F.G.); Niamh.OBoyle@tcd.ie (N.M.O.); nkeely@tcd.ie (N.O.K.); 2Department of Pharmaceutical Chemistry, College of Pharmacy, King Abdulaziz University, Jeddah 21589, Saudi Arabia; amelibary@kau.edu.sa; 3School of Biochemistry and Immunology, Trinity College Dublin, Trinity Biomedical Sciences Institute, 152-160 Pearse Street, Dublin 2 DO2R590, Ireland; FAYNED@tcd.ie (D.F.); seema.nathwani@outlook.com (S.M.N.); dzistrer@tcd.ie (D.M.Z.); 4School of Chemistry, Trinity College Dublin, Dublin 2 DO2R590, Ireland; TWAMLEYB@tcd.ie (B.T.); TMCCABE@tcd.ie (T.M.)

**Keywords:** Combretastatin A-4, β-lactam, 3-vinylazetidin-2-ones, antiproliferative activity, tubulin, antimitotic

## Abstract

Microtubule-targeted drugs are essential chemotherapeutic agents for various types of cancer. A series of 3-vinyl-β-lactams (2-azetidinones) were designed, synthesized and evaluated as potential tubulin polymerization inhibitors, and for their antiproliferative effects in breast cancer cells. These compounds showed potent activity in MCF-7 breast cancer cells with an IC_50_ value of 8 nM for compound **7s** 4-[3-Hydroxy-4-methoxyphenyl]-1-(3,4,5-trimethoxyphenyl)-3-vinylazetidin-2-one) which was comparable to the activity of Combretastatin A-4. Compound **7s** had minimal cytotoxicity against both non-tumorigenic HEK-293T cells and murine mammary epithelial cells. The compounds inhibited the polymerisation of tubulin in vitro with an 8.7-fold reduction in tubulin polymerization at 10 μM for compound **7s** and were shown to interact at the colchicine-binding site on tubulin, resulting in significant G2/M phase cell cycle arrest. Immunofluorescence staining of MCF-7 cells confirmed that β-lactam **7s** is targeting tubulin and resulted in mitotic catastrophe. A docking simulation indicated potential binding conformations for the 3-vinyl-β-lactam **7s** in the colchicine domain of tubulin. These compounds are promising candidates for development as antiproiferative microtubule-disrupting agents.

## 1. Introduction

Antimitotic agents such as taxol and the vinca alkaloids vinblastine and vincristine are a major class of drugs used clinically in the treatment of many cancers [1,2,3]. Microtubule-destabilizing agents (e.g., vinblastine) typically bind with tubulin at the vinca alkaloid site [4], while colchicine **1** exerts its biological effects at the intrasubunit interface within a tubulin dimer [5]. Stilbene-based compounds have attracted the attention of chemists and pharmacologists due to their many biological properties such as anticancer, antioxidant and anti-inflammatory activities, and are often used in traditional medicine for a variety of therapeutic effects [6]. The combretastatins are a group of stilbenes isolated from the South African bush willow tree *Combretum caffrum* [7], and are shown to have outstanding potency in binding to the colchicine-binding site of tubulin and thus inhibiting the formation of the mitotic spindle [8]. Combretastatin A-4 **2a** and Combretastatin A-1 **2c** demonstrate exceptionally potent antiproliferative activity against a range of human cancer cell lines (Figure 1) [7]. Additionally, antivascular effects are produced by these compounds in vivo [9,10]. Although some combretastatin compounds have progressed to clinical trials[11,12], there are major problems associated with combretastatins including poor water solubility and *cis/trans* isomerization during administration or storage, which results in an extensive loss of potency. Water soluble prodrugs such as the combretastatin phosphate CA-4P, (fosbretabulin) **2b** [13,14] are currently in clinical trials for advanced anaplastic thyroid carcinoma [15], ovarian cancer [16], and in combination with Bevacizumab for patients with advanced cancer [17]. Recently, the potential combination therapy of CA-4P and vincristine in the treatment of hepatocellular carcinoma was reported to show a beneficial effect in reducing doses of drugs with narrow therapeutic windows [18]. Ombrabulin is a serine prodrug whose derivatives display the same activity as CA-4 and has completed a phase III clinical trial for the treatment of advanced stage soft tissue sarcoma [19,20]. There is ongoing interest in the clinical development of combretastatin A1 diphosphate (OXi 4503) **2d** [21]. The structurally related benzophenones phenstatin **3a**, phenstatin phosphate **3b** [22] and the lignin podophyllotoxin **4** also destabilize microtubules [23].

Many heterocyclic scaffold structures have been introduced to replace the alkene of the stilbene structure of CA-4 and to provide conformational restriction by locking the stilbene in the *cis* configuration (Rings A and B) required for biological activity [24]. Small molecule tubulin polymerization inhibitors have been reported in which the *cis* double bond of CA-4 has been replaced by various heterocycles such as furan [25], indole[26,27], imidazole [28], isoxazole [29], triazole [30], tetrazole [31], benzoxepine [32], pyrazole [33], pyridine [34], benzimidazole [35] and related heterocycles [36]. While β-lactam antibiotics have occupied a central role in the treatment of pathogenic bacteria, the antiproliferative activity of compounds containing the β-lactam (azetidin-2-one) ring has also been investigated [37,38,39,40,41,42]. The synthesis and antitumour activity of a number of chiral β-lactam bridged CA-4 analogues have been reported [37,38]. Additional impetus for research efforts on β-lactam chemistry has been provided by the use of β-lactams as synthetic intermediates in organic synthesis [43].

We have previously investigated the antiproliferative and SERM (selective estrogen receptor modulator) activity of the azetidin-2-one(β-lactam) scaffold [44] and also demonstrated the effectiveness of 1,4-diarylazetidin-2-ones in breast cancer cell lines as tubulin targeting agents. [45,46]. These compounds also demonstrated both anti-angiogenic effects in MDA-MB-231 breast adenocarcinoma cells. In addition, we established that these compounds inhibited the migration of MDA-MB-231 cells indicating a potential anti-metastatic function for these compounds [47]. To further our understanding of the antiproliferative activity of these compounds, we wished to investigate the design, synthesis and evaluation of a series of azetidin-2-ones containing a vinyl substituent at C3 of the azetidin-2-one ring, and to explore the effect of this hydrophobic substituent on the biological activity of these compounds in which the *cis* configuration (Rings A and B) is locked into the azetidin-2-one ring structure. The introduction of this vinyl substituent at C-3 also allowed us to examine further chemical transformations of the alkene, and to determine structure-activity relationships for the series. On this basis, we now aimed to investigate a new series of novel 3-vinylazetidinones compounds with an improved biochemical profile particularly in triple negative breast cancer for potential development in preclinical study of breast cancer as tubulin destabilising agents. Therefore, we focused our efforts on the preparation of a library of 1,4-diarylazetidin-2-ones which contain a vinyl substituent at C-3. The synthesis of phosphate esters and amino acid amide type prodrugs of the most potent 1,4-diarylazetidin-2-ones were examined, together with the antiproliferative and tubulin targeting effects.

## 2. Results and Discussion

### 2.1. Chemistry: Synthesis of β-lactams

There are many synthetic routes available for the construction of the β-lactam ring [43,48]. The choice of route depends on the structural features required in the final product. In the present work, the Staudinger reaction between an imine and a ketene was chosen for the formation of the β-lactam ring because of its ease of use, adaptability for use with structurally diverse imines and acid chlorides, and readily available starting materials. A series of analogues with a variety of substituents at C4 of the β-lactam ring B was synthesized from the appropriate imines. The preparation of the Schiff bases **5a**–**5r** was achieved by the condensation of the appropriately substituted benzaldehyde with the 3,4,5-trimethoxyaniline in ethanol in the presence of a catalytic amount of sulphuric acid, (Scheme 1). The 3,4,5-trimethoxy substituted A-Ring of CA-4 plays an important role in inhibiting tubulin polymerisation, and is confirmed in the docking of CA-4 in tubulin [49]. The substituents located at the para-position of C-4 aryl Ring B included halogens (compounds **5a**–**5c**), nitro (**5d**), dimethylamino (**5e**), methyl (**5g**), alkoxyl (**5h**–**5j**), phenoxy (**5k**), benzyloxy (**5l**), nitrile (**5q**) and thiomethyl (**5r**) together with naphthyl (compounds **5m** and **5n**). **5s** was similarly obtained by reaction of 4-methoxybenzaldehyde with 3,5-dimethoxyaniline. For the synthesis of β-lactam derivatives with a phenolic hydroxy group to mimic Ring B of CA-4, it was necessary to use the benzyl ether **5l** and *tert*-butyldimethylsilyl ether **5o**. A further series of Schiff bases (**6a**–**6k**) was obtained from 3,4,5-trimethoxybenzaldehyde with appropriate anilines using the same procedure as above, (Scheme 1). An example of the crystal structure of the imine **6k** is displayed in Figure 2, showing the *E* configuration of the imine N1-C2 bond (bond length 1.278(2) Å) (Table 1).

A series of novel β-lactams (**7a**–**7r**) was obtained by reaction of imines **5a**–**5r** with crotonyl chloride using Staudinger reaction conditions requiring the slow addition of a solution of the appropriate acid chloride to a refluxing solution of imine and TEA, (Scheme 2) [50,51]. One enantiomer is illustrated in each case and products are obtained as a racemic mixture. β-Lactam (**7s**) containing the required Ring B phenolic group of CA-4 was successfully synthesised from the silyl ether imine **5o** and crotonyl chloride to afford the silyl ether β-lactam **7o** which was deprotected in situ by treatment with tBAF to yield the phenol **7s** (Scheme 2). This series of compounds **7a**–**7r** differ only in the substituent pattern of aryl ring at C-4 of the β-lactam ring B.

Many potentially useful CA-4 derivatives contain the amino substituent replacing the phenol on ring B and have shown interesting biochemical activity[52]. We were interested in the preparation of β-lactam CA-4 type compounds containing an amino substituent in Ring B, and the subsequent conversion to a water-soluble prodrug by conjugation with an amino acid. The nitro containing C-3-vinyl-β-lactam **7p** was successfully reduced to the amino product **7t** using zinc dust in the presence of acetic acid (Scheme 2). To investigate the effect of replacement of the 3,4,5-trimethoxy ring A of CA-4 with 3,5-dimethoxy substituted ring A, the β-lactam **7u** was prepared in a similar route from the imine **5s**. Tripodi et al. reported that 3,5-dimethoxy substituted ring A compounds demonstrated comparable activity to the β-lactam compounds containing the 3,4,5-trimethoxy ring A of CA-4 [53]. A further series of β-lactam compounds (**8a**–**8k**), was also prepared containing the 3,4,5-trimethoxyphenyl substituent (Ring A of the Combretastatin A-4) at C-4 position, (Scheme 2).

The products of the Staudinger reaction with imines and crotonyl chloride show IR absorptions at approximately ν 1750 cm^−1^ characteristic of the carbonyl group of the β-lactam ring. All of the β-lactams were obtained with exclusively *trans* stereochemistry, with coupling constants of 1–3 Hz for the β-lactam ring protons (e.g., for compound **7s**, H-4 is identified as a doublet δ 4.69, *J*_3,4_ = 2.52 Hz). Coupling constants of 5–6 Hz are usually observed for β-lactams with *cis* stereochemistry [46].

Subsequent to our initial biochemical evaluation of the 3-vinyl-β-lactam CA-4 analogues, a further series of 3-substituted β-lactams was prepared from 3-unsubstituted β-lactams by aldol type reaction with a suitable electrophile [54,55]. We were particularly interested in the introduction of modified alkene substituents at C-3, due to the exceptional biochemical activity displayed by the 3-vinyl β-lactam **7s**. Lithium enolates of 3-unsubstituted β-lactams **9a** and **9b** were reacted with selected aldehydes and ketones to provide alcohol products **10a**–**10i**, (Scheme 3). The β-lactams **9a** and **9b** were obtained via the Reformatsky reaction of ethyl bromoacetate with imines **5h** and **5o** using microwave conditions. Treatment of **9b** with tBAF afforded the phenol **9c**. Similarly, for the preparation of compounds **10a**, **10c**, **10d**, **10f**, **10g**, **10h** the initially obtained *t*BDMS ether intermediate was subsequently deprotected in situ using *t*BAF to yield the desired phenolic product. The enolate chemistry is stereoselective, favouring *trans* stereochemistry for the products. The presence of a diastereomeric mixture for products is confirmed from the ^1^H NMR spectra (e.g., for **10h** where H-3 and H-4 appear as two sets of doublets, δ 3.20 and δ 4.83 respectively, with *J* = 2.4 Hz, ratio H_3_/H_4_ 1.14:1.00). To investigate the role of the alcohol group at C-5 in the biochemical activity of the products **10a**–**10h**, the alcohol **10i** was oxidised to the corresponding ketone **11** using pyridinium chlorochromate. An alternative route to **11** was identified where treatment of the 3-unsubstituted β-lactam **9b** with LDA followed by addition of acetyl chloride to gave the desired product **11** but only in low yields (11%) with the alcohol **10a** also isolated (22%), (Scheme 3).

To further investigate the role of the 3-vinyl substitution pattern in the biochemical activity of. β-lactams, a 3-ethylidene product **12** was investigated. The initial route attempted involved the chlorination of the alcohol **10i** using thionyl chloride followed by dehydrohalogentation with a suitable base such as DBU. However, a more successful method to give the 3-ethylidene β-lactams was the dehydration of the alcohol **10i** under Mitsunobu conditions and subsequent deprotection by treatment with *t*BAF to yield **12** in 63% yield overall, (Scheme 3). The Peterson olefination of 3-unsubstituted β-lactams has also been reported by Kano et al. as an alternative route to 3-ethylidene β-lactams [56], while the Mitsunobu reaction for the dehydration of alcohols has been described by Plantan et al. in the synthesis of a trinem β-lactamase inhibitor [57]. The product **12** was obtained as a mixture of *Z*/*E* isomers in a 1:1 ratio. The configuration of the separated isomers was determined by examining the chemical shifts associated with the C-6 methyl protons. The further downfield doublet signal (δ 2.05, *J* = 4.16 Hz) is more deshielded, and so is assigned to the *Z* isomer while the signal at δ 1.62, (*J* = 4.40 Hz) is assigned to the *E* isomer [51].

The introduction of a diol functionality at C-3 was now explored. The diol **13** was synthesised in 39% yield by the oxidation of the alkene **7s** with osmium tetroxide (Scheme 3). The ^1^H NMR spectrum for **13** clearly illustrates the formation of a diastereomeric product. H-3 appears as a pair of double doublets at δ 3.16 (0.7H) and δ 3.19 (0.3H) with coupling constants of 2.42 Hz and 5.55 Hz, while H-4 appears as two separate doublets at δ 4.90 (0.3H) and δ 5.00 (0.7H), *J* = 2.37 Hz.

The amino acid alanine was chosen for prodrug formation of the β-lactam **7t** [58]. The protected amino acid prodrug **14** was obtained from **7t** using the coupling agent DCC with HOBt in dry DMF (Scheme 4). The FMOC protecting group was easily removed from **14** by treatment with 2N sodium hydroxide over 24 h to afford the amino acid prodrug conjugate **15** (57%). Controlled esterification of the phenolic β-lactams **7s** and **9c** with dibenzyl phosphite using diisopropylethylamine and dimethylaminopyridine afforded dibenzyl phosphate β-lactams **16a** and **16b** respectively, (Scheme 4). The dimethyl and diethylethyl phosphate esters of compound **9c**, **16c** and **16d** respectively, were also prepared (Scheme 4). The phosphate **17a** was obtained by treatment of dibenzylphosphate ester **16a** with bromotrimethylsilane. Hydrogenation of the dibenzylphosphate ester **16b** with palladium/carbon catalyst removed the dibenzyl protecting groups and also reduced the double bond at C-3 position of the β-lactam ring to afford phosphate **17c**. For the preparation of compound **17b**, where removal of the benzyl protecting groups and retention of the double bond was required, reaction of the dibenzyl phosphate ester **16b** with bromotrimethylsilane was effective.

Preliminary stability studies of the representative β-lactam **7s** were carried out at acidic, neutral and basic conditions (pH 4, 7.4 and 9) and in plasma using HPLC. The half-life (t_½_) was determined to be greater than 24 h at pH 4, 7.4 and 9 and in plasma for compound **7s**. The phosphate esters **17b** and **17c** were also found to be stable over the range of pH and in plasma, with half-life (t_½_) determined to be greater than 24 h. The cleavage of phosphate prodrugs **17b** and **17c** was also investigated in whole blood. They were cleaved much more rapidly in whole blood (62% and 34% remaining after 6 h respectively) than in human plasma (94% and 92% remaining after 6 h respectively). Based on this stability study the β-lactam **7s** would be suitable for further development.

### 2.2. X-Ray Structural Study

The X-ray crystal structures of compounds **7h, 8i, 8k** and **8h** are displayed in Figure 3 and confirm the structural assignment. The crystal data for the compounds are shown in Table 1 and Table 2. For each compound the two aryl rings at N-1 and C-4 position are in a pseudo *cis* arrangement while the phenyl ring at C4 and the alkene group are on opposite sides of the β-lactam (*trans* configuration). The structure of the compounds **7h**, **8h**, **8i** and **8k** clearly demonstrated a non-coplanar configuration for rings A and B of the β-lactams, with the β-lactam ring providing a rigid scaffold. For compound **7h** even though both enantiomers are present, the compound crystallizes out in a chiral space group. The *trans* configuration of the aryl rings A and B at C-3 and C-4 is also evident. The dihedral angle H3/H4 is observed for compounds **7h**, **8h**, **8i** and **8k** respectively, which is consistent with the small *trans* coupling constant observed in the ^1^H NMR spectrum of 2.00 Hz, 2.52 Hz, 2.48 Hz and 2.44 Hz respectively for these compounds. The β-lactam C=O bond lengths are 1.209(3) Å, 1.214(3) Å, 1.2077(17) Å and 1.2077(17) Å for compounds **7h**, **8h**, **8i** and **8k** respectively, which is consistent with data previously reported for the carbonyl bond length of monocyclic β-lactams of 1.217(3) Å [59] and 1.207(2) Å [60]. The ring A/B torsional angles for compounds **7h**, **8h**, **8i** and **8k** were observed as −59.5°, 59.7°, −73.5° and −77.0° respectively; these values are significantly greater than those observed for the corresponding rings A/B in the DAMA-colchicine **1b** [5], Combretastatin A-4 **2a** [61] and related 4-arylcoumarin [62] as 53°, 55° and 48.3° respectively (Table 2). The azetidinone N1-C4 bond length was observed at 1.372(3) Å, 1.376(3) Å, 1.367(2) Å and 1.3767(18) Å for compounds **7h**, **8h**, **8i** and **8k** respectively, which compares with 1.334(4)Å reported for the alkene C=C of combretastatin A-4 [61]. The C26-C27 alkene bond length for **7h**, **8h**, **8i** and **8k** were observed at 1.303(3) Å, 1.3174 Å, 1.308(3) Å and 1.316(2) Å respectively, while the alkene C=C bond length for *iso*-combretastatin CA-4 has been reported as 1.329(3) Å [63]. The C-N bonds lengths in the β-lactam ring are unequal with N1-C4 bond lengths of 1.487(3) Å, 1.483(3) Å, 1.4774(19) Å and 1.4801(17) Å for compounds **7h**, **8h**, **8i** and **8k** respectively, compared to 1.372(3) Å, 1.376(3) Å, 1.367(2) Å and 1.3767(18) Å for the N1/C2 bond in compounds **7h**, **8h**, **8i** and **8k** respectively, indicating some degree of amide resonance [59].

### 2.3. Biological Results and Discussion

#### 2.3.1. In vitro Antiproliferative Activities

The synthesized compounds were first evaluated for their antiproliferative activity against the human breast cancer cell line MCF-7 and compared with CA-4 as a reference compound (IC_50_ = 3.9 nM) [64,65]. The results are shown in Table 3 (**7a**–**7n**, **7p**–**7t**, **8a**–**8k**), and Table 4 (**10a–h**, **11**–**13**, **15** and **17a–c**). All β-lactams were evaluated as the *trans* isomer. The most potent compounds were identified as **7s** and **7t**, with IC_50_ values of 8 νM and 17 nM respectively. Compound **7s** is a direct analogue of CA-4, while **7t** is the corresponding amino compound and this type of substitution has been demonstrated to confer potency in many CA-4 analogues [52]. Compounds having the methoxy, ethoxy and thiomethyl substituents at C-4 of Ring B displayed potent antiproliferative effects, with IC_50_ values of 20 nM, 37 nM and 51 nM respectively for compounds **7h**, **7i** and **7r** respectively. The halo substituted compounds, **7b** and **7c** and 4-methyl compound **7g** were less effective with IC_50_ values of 690 nM, 445 nM and 355 nM respectively. Selectivity in antiproliferative effect was demonstrated by the 1 and 2-naphthyl compounds **7m** (IC_50_ = 1.738 μM) and **7n** (IC_50_ = 68 nM). This result compares favourably with the naphthyl CA-4 analogues reported by Medarde et al. in which the 2-naphthalene ring directly replaces the Ring B of CA-4 [66].

The IC_50_ of compound **7u** containing the 3,5-dimethoxyphenyl Ring A was determined as 170 nM in MCF-7 cells, demonstrating retention of antiproliferative potency with slightly reduced activity compared to the 3,4,5-trimethoxy ring A substituted compound **7h**. This observation could infer that the *para*-methoxy aryl group is less important for activity and the 3,5-dimethoxyaryl substituted Ring A is favourable for interaction of the molecule with the colchicine binding site of tubulin [53]. Compounds **8a**–**8k** containing the 3,4,5-trimethoxyphenyl substituent (Ring A of CA-4) at the C-4 position were generally observed to have poorer antiproliferative activity than the corresponding compounds **7a**–**7t**, containing the 3,4,5-trimethoxyphenyl substituent (Ring A of the Combretastatin A-4) at the N-1 position, (Table 3). The exceptions were compounds **8a** (4-fluoro) and **8j** (4-NHCOCH_3_) with IC_50_ values of 1.066 μM and 4.024 μM respectively. The relative positions of the 3,4,5-trimethoxyphenyl Ring A and Ring B on the β-lactam ring at positions N-1 and C-4 have a significant effect on the antiproliferative activity of the compounds as we previously reported [45].

The effects of various structural modifications on the activity of the more potent 3-vinylazetidinones were next explored, (Table 4). The most potent compound in this series is the 3-styryl containing compound **10f**, with IC_50_ = 46 nM. The alcohol **10a** showed interesting activity (65 nM) while the introduction of an additional methyl group at C-5 to afford the alcohol **10d** resulted in reduced efficacy with IC_50_ = 544 nM. The diol **13** also proved noteworthy with IC_50_ = 69 nM. The 3-acetyl compound **11** and 3-ethylidene compound **12** resulted in similar antiproliferative effects (IC_50_ = 414 nM and 502 nM respectively). Additional compounds containing the hydroxyalkene substituent at C-3 (e.g., compounds **10b**, **10c**, **10e**, **10g**, **10h**) were found to be moderately active (IC_50_ values 288–570 nM).

The amino acid prodrug amide **15** was evaluated in MCF-7 breast cancer cells to determine if it retained any antiproliferative activity when compared with the parent compound **7t** which was extremely potent with IC_50_ = 17 nM. The IC_50_ for **15** (3.251 μM) was lower than expected; however metabolic activation in vivo may be required for the hydrolysis of the amide [67]. The phosphate esters **17a**–**17c** displayed impressive antiproliferative activity, with IC_50_ values of 22 nM, 27 nM and 21 nM respectively (Table 4). The IC_50_ values for the corresponding phenols **7s** and **9c** in MCF-7 cells are 8 nM and 17 nM respectively. Comparison of the 3-vinyl **17b** (IC_50_ = 27 nM) with the 3-ethyl **17c** (IC_50_ = 21 nM) indicated that introduction of the 3-vinyl or 3-ethyl substituent, together with the 3-unsubstituted **17a** (22 nM) retains potency and optimum activity. The potent activity displayed for the phosphate esters **17a**–**17c**, together with the predicted improvement in water solubility, indicate that these compounds are useful prodrugs for future development. Rapid in vivo dephosphorylation would be expected to occur for the β-lactam phosphates **17a–c** as observed for CA-4P [11].

Triple-negative breast cancers (TNBC) are characterised by the absence of estrogen receptors (ER-), progesterone receptors (PR-) and human epidermal growth factor receptor 2 (HER2-). TNBC does not respond to hormonal therapy (such as tamoxifen or aromatase inhibitors) or therapies that target HER2 receptors, such as Herceptin. Treatment options are limited leading to poor prognosis, as indicated by low 5-year survival rates. A number of the more potent compounds were evaluated in the triple negative MDA-MB-231 cell line (Table 5). Compound **7s** was the most effective of the series with an IC_50_ value of 10 nM. Compounds **7h**, **7t**, **17a**, **17b** and **17c** were also seen to be effective with IC_50_ values of 31 nM, 30 nM, 30 nM, 49 nM and 44 nM respectively, and compared favourably with the positive CA-4 (control IC_50_ = 43 nM) [34,63,68].

Compound **7h** was also evaluated in the triple-negative Hs578T breast cancer cell line and its isogenic subclone Hs578Ts(i)8 cells to examine the activity of β-lactams as CA-4 analogues and as anti-tubulin agents for metastasis. Hs578Ts(i)8 cells are 3-fold more invasive and 2.5-fold more migratory than the parental cell line (Hs578T). In addition, Hs578Ts(i)8 cells had 30% more CD44+/CD24-/low cells that could enhance the invasive properties but with a significantly increased capacity to proliferate, migrate and produce tumours in vivo in nude mice [69]. Compound **7h** exhibited an excellent anti-proliferative activity in Hs578T cells (IC_50_ 31 nM) and interestingly retained potency in invasive Hs578Ts(i)8 cells (IC_50_ 76 nM). The values for CA4 in these cells were 8 nM and 20 nM respectively. These results could indicate the ability of β-lactams as CA-4 analogues to inhibit tumour invasion and angiogenesis which are characteristic of tumour growth and metastasis. These β-lactam compounds may provide potential development leads for this subset of aggressive breast cancers. Compound **7s** was also evaluated in the leukemia cell lines HL-60 and K562 and was found to be extremely potent with IC_50_ values of 17 nM and 26 nM respectively, comparing favourable with CA-4 [IC_50_ values of 4 nM (HL-60) and 4 nM (K562)].

The novel compounds **7h**, **7s, 7t**, **17b** and **17c** were selected for further investigation based on analysis of their drug-like properties (Lipinski) from a Tier-1 profiling screen, together with predictions of blood brain barrier partition, permeability, plasma protein binding, metabolic stability and human intestinal absorption properties which confirmed that these compounds are moderately lipophilic-hydrophilic drugs and are suitable candidates for further investigation (Appendix A).

#### 2.3.2. Evaluation of β-Lactams in the NCI60 Cell Line Screen

A series of the more potent compounds **7h**, **7s**, **7t**, **17b** and **17c** were evaluated in the National Cancer Institute (NCI)/Division of Cancer Treatment and Diagnosis (DCTD)/Developmental Therapeutics Program (DTP) [70], in which the activity of each compound was determined using approximately 60 different cancer cell lines of diverse tumor origins. The results are summarized in Appendix A. The compounds were tested for inhibition of growth (GI_50_) and cytotoxicity (LC_50_) in the NCI panel of cancer cell lines and showed excellent broad-spectrum antiproliferative activity against tumor cell lines derived from leukemia, non-small-cell lung cancer, colon cancer, CNS cancer, melanoma, ovarian cancer, renal cancer, breast cancer and prostate cancer [71] using the sulforhodamine B (SRB) protein assay [72], (Appendix A). The NCI results confirmed our in-house evaluations in MCF-7 cells with GI_50_ values for compounds **7h**, **7s**, **7t**, **17b** and **17c** of 30.6, <10, <10, 39.4 and 25.1 nM respectively.

Compound **7s**, the most potent compound in our panel, demonstrated a mean GI_50_ value of 23 nM across all NCI cell lines tested. The GI_50_ values for **7s** were in the sub-micromolar range for each of the cell lines investigated, except for two cell lines (melanoma cell line UACC-257 and the breast cancer cell line T-47D). For compound **7s** the GI_50_ values obtained were below 10 nM for 28 of the cell lines investigated and below 40 nM in all but eight of the panel cell lines tested. Activity was demonstrated for compound **7s** against all of the non-small cell lung (GI_50_ value 85.5 - <10 nM), colon (GI_50_ value 429 - <10 nM), CNS (GI_50_ value 40.5 - <10 nM), ovarian (GI_50_ value 45.3 - <10 nM), prostate (GI_50_ value <10 nM) and renal (GI_50_ value 40.2 - <10 nM) cancer cell lines tested. The mean GI_50_ values over the full 60 cell line panel for compounds **7h**, **7t** and **17c** of 52, 48 and 73 nM respectively (see Appendix A) compares very favourably with the GI_50_ value for CA-4 of 99 nM.

LC_50_ values for compound **7s** were greater than 100 μM in all but three cell lines tested indicating minimal toxicity and the potential use of this compound for a wide range of therapeutic applications (Appendix A). A similar result was obtained for compound **7h** with LC_50_ values > 100 μM in all cell lines tested.

The NCI COMPARE algorithm allows a comparison of the activities of β-lactams **7h** and **7s** with compounds of a known mechanism of antiproliferative action in the NCI Standard Agents Database. Compounds **7h** and **7s** showed high correlation to tubulin targeting agents such as maytansine, rhizoxin and the clinically important anticancer drugs vincristine and vinblastine, (see Appendix A).

#### 2.3.3. Evaluation of Toxicity of 7s in Normal Murine Mammary Epithelial Cells

The cytotoxic effect of a selected number of 3-vinyl-β-lactams in MCF-7 cells at 10 μM concentration was initially determined in the lactate dehydrogenase (LDH) assay [73]. The 3-vinyl-β-lactams **7d**, **7h**, **7i**, **7q**, **7r** and **7u** resulted in low cytotoxicity with 7.2%, 2.4%, 8.5%, 4.5%, 4.5% and 3.5% cell death respectively while compounds **7s** and **7t** displayed increased cytotoxicity of 16.1% and 25% cell death in this assay. The 3-(1-hydroxyl-1-methylethyl) and 3-(1-hydroxy-1-phenylallyl) substituted compounds **10d** and **10f** resulted in 9.8% and 7.6% cell death respectively while cell death of 8.4% was obtained for the 3-ethylidene compound **12**. CA-4 was used as the positive control in this assay and resulted in 11.8% cell death at 10 μM concentration.

The cytotoxicity of the most potent compound **7s** on non-tumourigenic cell line HEK-293 (normal human embryonic kidney) was also investigated. We demonstrated an IC_50_ value greater than 5 µM in HEK-293T cells for **7s**. Cell viability of HEK-293T cells was significantly higher than MCF-7 cells at 10, 1 and 0.5 µM concentrations of compound **7s** (Figure 4A), demonstrating the lack of cellular toxicity of the compounds in these non-cancerous cells.

Further toxicity studies were carried out on the most potent compound β-lactam **7s** in primary cells (mouse mammary healthy epithelial cells) at two different cell concentrations (25,000 and 50,000 cells/mL), with CA4 as a positive control. The cells were harvested from mid- to late-pregnant CD-1 mice and were cultured as previously reported [74,75]. Both CA-4 [76] and **7s** were not cytotoxic at concentrations up to 10 μM in the NCI cell line panel (See Appendix A). The IC_50_ values for both compounds **7s** and CA-4 evaluated in normal murine mammary epithelial cells was greater than 10 μM which indicated a minimal toxicity for these compounds (Figure 4B). At both 25,000 cells/mL and 50,000 cells/mL and a concentration of 10 μM, CA-4 was lethal to the highest percentage of cells. The percentage of viable murine mammary epithelial cells at the IC_50_ value of each compound in MCF-7 cells (see Table 3) was calculated in order to give an estimation of the toxicity at this value. At 50,000 cells/mL, over 90% of cells were viable after 72 h for compound **7s**, (Figure 4B). At 25,000 cells/mL, the percentage of cells remaining viable after treatment with compound **7s** for 72 h was 93%, compared to 74% for CA-4. (Appendix A). These results indicate a favourable toxicity profile for **7s** in comparison to CA4. This provides further evidence, in addition to the NCI60 LC_50_ values for **7s**, that the β-lactam compound developed in this study is minimally toxic to cells that are not proliferating.

#### 2.3.4. Effect of β-Lactam 7s on Cell Cycle and Apoptosis

It is well recognised that tubulin destabilizing agents arrest the cell cycle in the G_2_/M phase due to cytoskeleton disruption and microtubule depolymeriztion. The effects of β-lactam **7s** on cell cycle events and induction of apoptosis in MCF-7 cells were next explored. Initial analysis by flow cytometry of propidium iodide stained MCF-7 cells showed G_2_M arrest at 24 h by compound **7s** [64% (10 nM) and 82% (100 nM)] (Figure 5C). A time dependent increase in the percentage of apoptotic cells (sub-G_0_G_1_) after 72 h (14% and 26% respectively for 10 nM and 100 nM concentration) was also evident compared to the vehicle control (6% at 72 h), (Figure 5A), with a corresponding decrease of cells in the G_0_–G_1_ phase of the cell cycle, (Figure 5C). The positive control CA-4 (100 nM) showed 52% of cells in G_2_M arrest at 48 h, and 9.4% in the sub-G_0_G_1_ population.

To characterize the mode of cell death induced by **7s** in MCF-7 cells, analysis of apoptosis was performed using propidium iodide (PI), which stains DNA and enters only dead cells, and annexin-V, which binds selectively to phosphatidyl serine (Figure 6). Dual staining for annexin-V and PI facilitates discrimination between live cells (annexin-V-/PI-), early apoptotic cells (annexin-V+/PI-), late apoptotic cells (annexin-V+/PI+) and necrotic cells (annexin-V-/PI+). Each concentration induced an accumulation of annexin-V positive cells when compared to the vehicle control (5%), Figure 6. About 13.6% of cells were found to be apoptotic (annexin-V positive) when treated with compound **7s** at 10 nM for 72 h. With an increase in concentration of **7s**, 31.9% of cells were found to be apoptotic at 100 nM. The positive control CA-4 (50 nM) resulted in 34.6% apoptotic cells. The observed effect of compound **7s** on cell cycle resulting in G_2_M arrest followed by apoptosis is typical of tubulin targeting compounds. However, we have previously reported that prolonged exposure of colon cancer cells CT-26, CaCo-2 and HT-29 to our structurally related 3-aryl-β-lactams induced autophagy [77]; it is possible that autophagy may be the cell death mechanism in the present case, because of the level of apoptosis observed.

#### 2.3.5. Tubulin Polymerization Studies

The effect of selected β-lactam CA-4 compounds (**7h**, **7i**, **7s**, **7t**) which demonstrated the most potent antiproliferative effects in vitro was assessed on the assembly of purified bovine tubulin. CA-4 which effectively inhibits the assembly of tubulin was used as a positive control, while paclitaxel was used to demonstrate effective tubulin polymerization. Tubulin polymerization was determined for compounds **7h**, **7i** and **7t** at 10 μM for 30 min and compound **7s** at 1, 5 and 10 μM for 60 min by measuring the increase in absorbance at 340 nm, (Figure 7A,B) [78]. The degree of light scattering by microtubules is proportional to their degree of polymerization. For the paclitaxel control the v_max_ was found to be 89.4 mOD/min. The v_max_ value provides a sensitive indication of the tubulin/ligand interactions for the tubulin polymerization. The most potent antiproliferative compound **7s** (10 μM) demonstrated a significant 8.7-fold reduction in v_max_ value while exposure to CA-4 (10 μM) brings about a 5.28-fold reduction in the v_max_ value. Compound **7s** compares very favourably to CA-4 in this respect. These effects are in good agreement with the antiproliferative data recorded for both CA-4 (IC_50_ = 4.2 nM) and **7s** (IC_50_ = 8 nM) in the MCF-7 cell line. The v_max_ value for compounds **7h**, **7i** and **7t** was determined as 3.43, 3.84 and 0.92 mOD/min respectively, together with the fold-reduction in the v_max_ values of 2.45, 2.19 and 9.15 respectively for the tubulin polymerization with reference to ethanol control. These results confirm that the molecular target of these antiproliferative 3-vinyl-β-lactams is tubulin and that they are microtubule-destabilising agents.

The dose-dependent effect of **7s** on tubulin polymerization is illustrated in Figure 7. Exposure of the tubulin to 10 μM, 5 μM and 1 μM of **7s** resulted in a dose-dependent fold reduction of v_max_ of 8.70, 7.31 and 2.61 respectively while the IC_50_ value for **7s** for the inhibition of polymerization was calculated to be 1.37 μM, Figure 7. Taken together, these results demonstrate that for these novel β-lactam containing CA4 analogues, antiproliferative activity against the MCF-7 cell line and the inhibition of tubulin polymerization are closely related. It has also been shown that the most potent antiproliferative compound synthesised (**7s**) inhibits tubulin polymerization to a greater extent than CA-4.

#### 2.3.6. Immunofluorescence Microscopy

Alterations in the microtubule network induced by β-lactam **7s** in MCF-7 cells were investigated using immunofluorescence and confocal microscopy (Figure 8). A well organised microtubular network was observed in MCF-7 control cells when stained with α-tubulin mAb (Figure 8) and in untreated cells (data not shown). Formation of microtubule bundles and pseudo asters was demonstrated for cells when exposed to paclitaxel (a microtubule-stabilising agent), Figure 8 [79]. A complete loss of microtubule formation was induced in cells exposed to CA-4 or β-lactam **7s** for 16 h. This effect is consistent with depolymerised microtubules. Following treatment with CA-4 or β-lactam **7s**, MCF-7 cells were observed to contain multiple micronuclei. Mitotic catastrophe resulting from premature or inappropriate entry of cells into mitosis is a type of programmed cell death in response to DNA damage, and is characterised by multinucleated cells [80]. CA-4 induced mitotic catastrophe has also been reported in non-small cell lung cancer cells [81,82], human endothelial cells (HUVEC) [83], human lung carcinoma cells (H460) [83] and human breast cancer cells (MCF-7) [84]. Taken together with the effects demonstrated above in Section 2.3.5 on the inhibition of polymerisation of isolated tubulin, the confocal imaging results confirm that β-lactam **7s** is targeting tubulin.

#### 2.3.7. Interaction of β-Lactam 7s with Colchicine Binding Site of Tubulin

The binding of the lead compound **7s** to the colchicine binding site of tubulin was confirmed in a whole cell-based assay. *N*,*N*′-ethylene-bis(iodoacetamide) (EBI) is an alkylating agent that cross-links cysteine residues 239 and 354 in the colchicine-binding site of tubulin to form the β-tubulin-EBI adduct that migrates faster than β-tubulin [85,86], and is detected by Western blotting. However, when the MCF-7 cells are pre-treated with colchicine or a colchicine-site ligand such as CA-4, the formation of the β-tubulin-EBI adduct is prevented. The MCF-7 cells were initially treated with selected β-lactam **7s** (10 μM) or CA- 4 for 2 h, then followed by addition of EBI for a further 1.5 h (Figure 9). The presence of the β-tubulin-EBI adduct was demonstrated for the control samples (no drug) at a lower position on the gel, indicating that EBI has cross-linked Cys239 and Cys354 on β-tubulin. When the cells are treated with β-lactam **7s** and CA-4, the EBI adduct formation is inhibited, indicating that **7s** is interacting with tubulin at the colchicine site of tubulin.

### 2.4. Molecular Modelling Studies

The 3-vinyl-β-lactam compound **7s** represents the most potent compound synthesised in the study with IC_50_ value of 8 nM in MCF-7 breast cancer cells. The tubulin binding and immunofluorescence studies of 3-vinyl-β-lactam **7s** have demonstrated that the colchicine binding site of tubulin is the target for the compound. Flexible alignment of compound **7s** with CA-4 resulted in a good degree of overlap between the trimethoxyphenyl rings (Ring A) and the phenolic hydroxyl group of ring B (Figure 10A). The energy minimised structure of compound **7s** demonstrates the inter-atomic distances of the oxygens of the methoxy groups of ring A and ring B as 9.17 Å, which is similar to that calculated for CA-4 (9.27 Å).

The X-ray structure of CA-4 co-crystallised with tubulin has been determined suggesting that *cis*-CA-4 inhibits tubulin polymerization by preventing the transition from curved to straight tubulin [49]. The X-ray structure of *cis* and *trans* stereoisomers of a 3-methyl-1,4-diarylazetidinone [87] co-crystallised with tubulin was reported by Zhou et al. [37,38]. In the present study the potential interaction of our novel synthesised 3-vinyl-β-lactams with the colchicine binding site of tubulin, a series of docking calculations using MOE 2018.0101 [88] was undertaken on both the 3*S*/4*R* and 3*R*/4*S* enantiomers of the β–lactams **7s** and **7t** using the tubulin co-crystallised with DAMA-colchicine X-ray crystal structure (PDB entry 1SA0) [5]. Only results for the 3*S*/4*R* studies will be discussed as these stereoisomers were more highly ranked than the 3*R*/4*S* enantiomer and this is also supported by the crystallographic evidence [37,38]. The 3*S*/4*R* enantiomers of the hydroxyl **7s** and amino **7t** substituted analogues overlay their B-rings on the C-ring of DAMA-colchicine, collocate the trimethoxyphenyl substituents, overlap the 3-hydroxyl/amino groups on the DAMA-colchicine carbonyl oxygen atom and form HBA interactions with Lys β352 as shown in Figure 10B and 10C. The 3,4,5-trimethoxyphenyl groups of all analogues are able to make favourable van der Waals contacts within the lower subpocket delineated by Val β318 and Cys β241. The β-lactam carbonyl oxygen atom can make an HBA interaction with the backbone amine of Asp β251 for both analogues. For both compounds, the *trans* geometry at C3/C4 facilitates a more favourable interaction of rings A and B with the residues of the β-tubulin colchicine binding site. Protein-ligand interactions for **7s** are illustrated in Figure 11. The enantioselective synthesis of **7s** and **7t** are in progress which will provide the optimum configuration of these compounds to be determined for biological activity.

## 3. Conclusions

We have developed an interesting series of 3-vinylazetidinones which selectively modulate the activity of the tubulin protein, resulting in significant cytotoxicity to cancer cells and minimum cytotoxic effects to normal cells. Molecular modelling studies indicated that these compounds could interact with the colchicine binding site of tubulin, and consequently disrupt tubulin polymerization. X-Ray crystallographic studies confirmed that the torsional angle between Ring A and Ring B of the β-lactam was similar to CA-4 and was important in maintaining antiproliferative and tubulin disrupting activity. Biochemical evaluation of these compounds coupled with a molecular modeling study contributes to our understanding of the attributes of the 3-vinylazetidinones such as **7s** and **7t** that result in exceptional antiproliferative activity and dose-dependent microtubule assembly inhibition. Analysis of DNA content by flow cytometry demonstrated that the cells were arrested in the G_2_/M phase; induction of apoptosis was confirmed by an increase in the sub-G_0_G_1_ population, which was confirmed by Annexin-V staining. Immunofluorescence staining with α-tubulin antibodies in MCF-7 cells demonstrated disorder and fragmentation of the microtubule network and disruption of mitotic spindle formation. The phosphate prodrugs **17a–c** were found to retain antitumour potency. The potent antiproliferative activity of the 3-vinylazetidinones **7s** and **7t** in breast cancer cells MCF-7 and notably in the triple negative MDA-MB-231 cell line reported in the present study compare very favourably with examples from the related series of 3-arylazetidinone compounds previously reported by our research group [45]. Vinyl substitution at C-3 of these azetidinones also results potent tubulin destabilizing effects in these derivatives of combretastatin A-4.

In summary, these novel 3-vinyl-β-lactam analogues of CA-4 which we now report show potent antiproliferative effects in preliminary in vitro investigations on MCF-7 and MDA-MB231 breast cancer cells. Further studies to establish the long-term effect of these compounds on cancer cell growth, migration and the potential vascular disrupting effects of these molecules are ongoing.

## 4. Experimental Section

### 4.1. Chemistry

All reagents were commercially available and were used without further purification unless otherwise indicated. Tetrahydrofuran (THF) was distilled immediately prior to use from Na/Benzophenone under a slight positive pressure of nitrogen, toluene was dried by distillation from sodium and stored on activated molecular sieves (4Å) and dichloromethane was dried by distillation from calcium hydride prior to use. Uncorrected melting points were measured on a Gallenkamp SMP 11 melting point apparatus. Infra-red (IR) spectra were recorded as thin film on NaCl plates, or as potassium bromide discs on a Perkin Elmer FT-IR Spectum 100 spectrometer, (PerkinElmer Inc., 940 Winter Street, Waltham, MA, USA). ^1^H and ^13^C nuclear magnetic resonance (NMR) spectra were recorded at 27 °C on a Brucker Avance DPX 400 spectrometer (Bruker, 40 Manning Road, Billerica, MA, USA), (400.13 MHz, ^1^H; 100.61 MHz, ^13^C) at 20 °C in either CDCl_3_ (internal standard tetramethylsilane TMS) or CD_3_OD by Dr. John O’Brien and Dr. Manuel Ruether in the School of Chemistry, Trinity College Dublin. For CDCl_3_, ^1^H-NMR spectra were assigned relative to the TMS peak at δ 0.00 ppm and ^13^C-NMR spectra were assigned relative to the middle CDCl_3_ triplet at δ 77.00 ppm. For CD_3_OD, ^1^H and ^13^C-NMR spectra were assigned relative to the centre peaks of the CD_3_OD multiplets at δ 3.30 and 49.00 ppm respectively. Electrospray ionisation mass spectrometry (ESI-MS) was performed in the positive ion mode on a liquid chromatography time-of-flight (TOF) mass spectrometer (Micromass LCT, Waters Ltd., Manchester, UK) equipped with electrospray ionization (ES) interface operated in the positive ion mode at the High Resolution Mass Spectrometry Laboratory by Mr. Brian Talbot in the School of Pharmacy and Pharmaceutical Sciences, Trinity College Dublin and Dr. Martin Feeney in the School of Chemistry, Trinity College Dublin. Mass measurement accuracies of < ±5 ppm were obtained. Low resolution mass spectra (LRMS) were acquired on a Hewlett-Packard 5973 MSD GC-MS system in electron impact (EI) mode, (Hewlett-Packard, 6280 America Center, San Jose, CA, USA). R_f_ values are quoted for thin layer chromatography on silica gel Merck F-254 plates, unless otherwise stated, (Merck, 2000 Galloping Hill Road, Kenilworth, NJ, USA). Flash column chromatography was carried out on Merck Kieselgel 60 (particle size 0.040–0.063 mm). Chromatographic separations were also carried out on Biotage SP4 instrument, (Biotage AB, Box 8, Uppsala, Sweden). All products isolated were homogenous on TLC. Analytical high-performance liquid chromatography (HPLC) to determine the purity of the final compounds was performed using a Waters 2487 Dual Wavelength Absorbance detector, a Waters 1525 binary HPLC pump, a Waters In-Line Degasser AF and a Waters 717plus Autosampler, (Waters, 34 Maple St, Milford, MA, USA). The column used was a Varian Pursuit XRs C18 reverse phase 150 × 4.6 mm chromatography column (Agilent Technologies, 5301 Stevens Creek Blvd, Santa Clara, CA, USA). Samples were detected using a wavelength of 254 nm. Imines **5a** [87], **5b** [89], **5c** [46], **5d** [46], **5e** [84], **5f** [90], **5g** [91], **5h** [45], **5i** [84], **5m** [46], **5n** [84], **5o** [45], **5p** [45], **5q** [46], **5r** [46], **5s** [53], **6a** [92], **6b** [93], **6c** [94], **6f** [45], **6g** [95], **6h** [96], **6i** [45], **6j** [97], **6k** [96] were prepared as previously reported (See Appendix A).

#### 4.1.1. 3-(tert-Butyldimethylsilyloxy)-4-methoxybenzaldehyde

To a solution of 3-hydroxy-4-methoxybenzaldehyde (20 mmol) and *tert*-butyl-dimethylsilylchloride (24 mmol) in dry CH_2_Cl_2_ (60 mL) under a nitrogen atmosphere, 1,8-diazabicyclo[5.4.0]undec-7-ene (DBU) (32 mmol) was added dropwise via syringe. The resulting mixture was stirred at room temperature under a nitrogen atmosphere until reaction was complete on thin layer chromatography. The solution was then diluted with CH_2_Cl_2_ (80 mL) and washed successively with water (60 mL), 0.1M HCl (60 mL) and saturated aqueous NaHCO_3_ (60 mL), retaining the organic layer each time, before drying over anhydrous Na_2_SO_4_. The solvent was removed under reduced pressure to yield the protected benzaldehyde, yield 82% [45]. IR (NaCl, film) ν_max_: 1692 (C=O) cm^−1^. ^1^H NMR (400 MHz, CDCl_3_): δ 0.19 (s, 6H), 1.02 (s, 9H), 3.91 (s, 3H), 6.97 (d, *J* = 8.56 Hz, 1H), 7.38 (d, *J* = 2.00 Hz, 1H), 7.48–7.51 (m, 1H), 9.84 (s, 1H). ^13^C NMR (100 MHz, CDCl_3_): δ -5.07, 17.97, 25.19 (OTBDMS), 55.13, 110.71, 119.63, 125.82, 129.75, 145.12, 156.16, 190.48. HRMS: found 266.1349 (M^+^); C_14_H_22_O_3_Si requires 266.1338.

#### 4.1.2. General Method I: Preparation of Imines **5a**–**5s**, **6a**–**6k**

The appropriately substituted benzaldehyde (10 mmol) and corresponding substituted aniline (10 mmol) were heated reflux in ethanol (40 mL) for 4 h with a catalytic amount of concentrated sulphuric acid. The volume of reaction was then reduced to approximately 10 mL in vacuo. The Schiff base precipitated from solution upon standing at room temperature overnight. The solid product obtained was filtered and purified by recrystallisation from ethanol.

##### (*E*)-1-(4-Butoxyphenyl)-*N*-(3,4,5-trimethoxyphenyl)methanimine (**5j**)

Preparation as described above from 4-butoxybenzaldehyde and 3,4,5-trimethoxyaniline. The product was obtained as pale yellow solid, yield 67%, Mp: 107–108 °C. IR (KBr) ν_max_: 1607 (C=N) cm^−1^. ^1^H NMR (400 MHz, CDCl_3_): δ 1.01 (t, *J* = 7.26Hz, 3H), 1.51–1.56 (m, 2H), 1.79–1.84 (m, 2H), 3.88 (s, 3H), 3.92 (s, 6H), 4.06 (t, *J* = 6.48 Hz, 2H), 6.42 (br s, 2H), 7.00 (d, *J* = 8.52 Hz, 2H), 7.87 (br s, 2H), 8.41 (s, 1H). ^13^C NMR (100 MHz, CDCl_3_): δ 13.87, 19.24, 31.22, 56.13, 61.04, 67.93, 98.11, 114.76, 128.57, 130.65, 136.12, 148.17, 153.56, 159.24, 162.09. HRMS: found 344.1859 (M^+^+H); C_20_H_26_NO_4_ requires 344.1862.

##### (*E*)-1-(4-Phenoxyphenyl)-*N*-(3,4,5-trimethoxyphenyl)methanimine (**5k**)

Preparation as described above from 4-phenoxybenzaldehyde and 3,4,5-trimethoxyaniline. The product was obtained as pale yellow solid, yield 74%, Mp 86–88 °C. IR (KBr) ν_max_: 1631 (C=N) cm^−1^. ^1^H NMR (400 MHz, CDCl_3_): δ 3.88 (s, 3H), 3.92 (s, 6H), 6.51 (s, 2H), 7.08–7.10 (m, 5H), 7.39–7.41 (m, 2H), 7.89 (d, *J* = 7.52 Hz, 2H), 8.45 (s, 1H). ^13^C NMR (100 MHz, CDCl_3_): δ 55.68, 60.58, 97.68, 117.80, 119.34, 123.78, 124.51, 129.53, 130.10, 135.83, 147.48, 153.12, 155.57, 158.32, 160.06. HRMS: found 364.1534 (M^+^+H); C_22_H_22_NO_4_ requires 364.1549.

##### (*E*)-1-(4-(Benzyloxy)phenyl)-*N*-(3,4,5-trimethoxyphenyl)methanimine (**5l**)

Preparation as described above from 4-(benzyloxy)benzaldehyde and 3,4,5-trimethoxyaniline. The product was obtained as colourless solid, yield 79%, Mp 113–115 °C. IR (KBr) ν_max_: 1623 (C=N) cm^−1^. ^1^H NMR (400 MHz, CDCl_3_): δ 3.88 (s, 3H), 3.92 (s, 6H), 5.17 (s, 2H), 6.51 (s, 2H), 7.09 (d, *J* = 7.84 Hz, 2H), 7.41–7.46 (m, 5H), 7.89 (d, *J* = 7.84 Hz, 2H), 8.42 (s, 1H). ^13^C NMR (100 MHz, CDCl_3_): δ 55.67, 60.58, 69.66, 97.65, 114.69, 127.90, 127.06, 127.75, 128.23, 128.30, 128.66, 130.16, 135.97, 147.19, 153.10, 158.63, 161.07. HRMS: found 378.1713 (M^+^+H); C_23_H_24_NO_4_ requires 378.1705.

##### (*E*)-*N*-(4-Nitrophenyl)-1-(3,4,5-trimethoxyphenyl)methanimine (**6d**)

Preparation as described above from 3,4,5-trimethoxybenzaldehyde and 4-nitroaniline. The product was obtained as yellow solid, yield 72%, Mp 161–162 °C. IR (KBr) ν_max_: 1627 (C=N) cm^−1^. ^1^H NMR (400 MHz, CDCl_3_): δ 3.96 (s, 3H), 3.97 (s, 6H), 7.20 (s, 2H), 7.27 (d, *J* = 8.52 Hz, 2H), 8.28 (d, *J* = 9.04 Hz, 2H), 8.34 (s, 1H). ^13^C NMR (100 MHz, CDCl_3_): δ 55.82, 55.85, 60.62, 105.93, 120.86, 124.62, 130.12, 141.48, 144.97, 153.12, 155.56, 161.70. HRMS: found 317.1135 (M^+^+H); C_16_H_17_N_2_O_5_ requires 317.1137.

##### (*E*)-*N*-(4-(tert-Butyl)phenyl)-1-(3,4,5-trimethoxyphenyl)methanimine (**6e**)

Preparation as described above from 3,4,5-trimethoxybenzaldehyde and 4-(*tert*-butyl)aniline. The product was obtained as red solid, yield 100%, Mp 81–82 °C. IR (KBr) ν_max_: 1623 (C=N) cm^−1^. ^1^H NMR (400 MHz, CDCl_3_): δ 1.37 (s, 9H), 3.94 (s, 3H), 3.97 (s, 6H), 7.15–7.21 (m, 4H), 7.44 (d, *J* = 8.52 Hz, 2H), 8.40 (s, 1H). ^13^C NMR (100 MHz, CDCl_3_): δ 30.98, 34.07, 55.82, 60.55, 105.29, 120.10, 125.61, 131.31, 140.41, 148.59, 148.70, 153.05, 158.78. HRMS: found 328.1899 (M^+^+H); C_20_H_26_NO_3_ requires 328.1913.

#### 4.1.3. General method II: Preparation of 2-azetidinones **7a**–**7u**, **8a**–**8k**

To a stirring, refluxing solution of the imine (5 mmol) and triethylamine (6 mmol) in anhydrous dichloromethane (40 mL), a solution of crotonyl chloride (6 mmol) in anhydrous dichloromethane (10 mL) was injected dropwise through a rubber septum over 45 min under nitrogen. The reaction was heated at reflux for 5 h and stirred at room temperature overnight, continuously under nitrogen. The reaction mixture coled and washed with water (2 × 100 mL), with the organic layer being retained each time. The reaction was dried over anhydrous sodium sulfate and the solvent was then removed under reduced pressure. The crude product was purified by flash chromatography over silica gel (eluent: *n*-hexane: ethyl acetate, 4:1).

##### 4-(4-Fluorophenyl)-1-(3,4,5-trimethoxyphenyl)-3-vinylazetidin-2-one (**7a**)

Preparation as described above from crotonyl chloride and (4-fluorobenzylidene)-3,4,5-trimethoxyphenylamine (**5a**). The product was obtained as yellow solid, yield 36%, Mp 147–149 °C. IR (KBr) ν_max_: 1749 (C=O) cm^−1^. ^1^H NMR (400 MHz, CDCl_3_): δ 3.73 (s, 6H), 3.74–3.75 (m, 1H), 3.78 (s, 3H), 4.78 (d, *J* = 2.52 Hz, 1H), 5.35–5.43 (m, 2H), 5.99–6.08 (m, 1H), 6.53 (s, 2H), 7.09–7.13 (m, 2H), 7.36–7.38 (m, 2H). ^13^C NMR (100 MHz, CDCl_3_): δ 55.57, 60.51, 60.51, 63.51, 94.20, 115.76, 115.97, 119.74, 127.18, 127.25, 129.81, 132.63, 133.16, 134.09, 153.10, 161.10, 164.56. HRMS: found 356.1303 (M^+^-H); C_20_H_19_FNO_4_ requires 356.1298.

##### 4-(4-Chlorophenyl)-1-(3,4,5-trimethoxyphenyl)-3-vinylazetidin-2-one (**7b**)

Preparation as described above from crotonyl chloride and (4-chlorobenzylidene)-(3,4,5-trimethoxyphenyl)amine (**5b**) to afford the product as a yellow solid, yield 30%, Mp 104 °C. IR (KBr) ν_max_: 1754 (C=O) cm^−1^. ^1^H NMR (400 MHz, CDCl_3_): δ 3.70–3.71 (m, 1H), 3.72 (s, 6H), 3.77 (s, 3H), 4.77 (d, *J* = 2.48 Hz, 1H), 5.33–5.41 (m, 2H), 5.97–6.06 (m, 1H), 6.51 (s, 2H), 7.32 (d, *J* = 6.52 Hz, 2H), 7.38 (d, *J* = 8.56 Hz, 2H). ^13^C NMR (100 MHz, CDCl_3_): δ 56.06, 60.88, 60.95, 63.93, 94.64, 120.30, 127.28, 129.50, 130.19, 133.55, 134.54, 134.59, 135.89, 153.58, 164.89. HRMS: found 372.1017 (M^+^-H); C_20_H_19_^35^ClNO_4_ requires 372.1003.

##### 4-(4-Bromophenyl)-1-(3,4,5-trimethoxyphenyl)-3-vinylazetidin-2-one (**7c**)

Preparation as described above from crotonyl chloride and (4-bromobenzylidene)-3,4,5-trimethoxyphenylamine (**5c**) to afford the product as a brown oil, yield 48%. IR (NaCl) ν_max_: 1751 (C=O) cm^−1^. ^1^H NMR (400 MHz, CDCl_3_): δ 3.71–3.72 (m, 1H), 3.74 (s, 6H), 3.79 (s, 3H), 4.76 (d, *J* = 2.52 Hz, 1H), 5.35–5.43 (m, 2H), 5.99–6.08 (m, 1H), 6.53 (s, 2H), 7.27 (d, *J* = 8.52 Hz, 2H), 7.55 (d, *J* = 8.52 Hz, 2H). ^13^C NMR (100 MHz, CDCl_3_): δ 55.63, 60.50, 60.52, 63.43, 94.19, 119.88, 122.19, 127.08, 129.69, 132.01, 133.06, 135.95, 139.98, 153.13, 164.42. HRMS: found 418.0643 (M^+^+H); C_20_H_21_^79^BrNO_4_ requires 418.0654.

##### 4-(4-Nitrophenyl)-1-(3,4,5-trimethoxyphenyl)-3-vinylazetidin-2-one (**7d**)

Preparation as described above from crotonyl chloride and (4-nitrobenzylidene)-3,4,5-trimethoxyphenylamine (**5d**) to afford the product as a brown solid, yield 28%, Mp 132–133 °C. IR (KBr) ν_max_: 1754 (C=O) cm^−1^. ^1^H NMR (400 MHz, CDCl_3_): δ 3.73 (s, 6H), 3.75–3.76 (m, 1H), 3.78 (s, 3H), 4.92 (d, *J* = 2.48 Hz, 1H), 5.39–5.45 (m, 2H), 6.01–6.06 (m, 1H), 6.50 (s, 2H), 7.56 (d, *J* = 9.04 Hz, 2H), 8.28 (d, *J* = 9.04 Hz, 2H). ^13^C NMR (100 MHz, CDCl_3_): δ 55.68, 60.03, 60.52, 63.53, 94.16, 120.46, 124.14, 126.26, 129.24, 132.75, 134.44, 144.28, 147.61, 153.27, 163.83. HRMS: found 385.1389 (M^+^+H); C_20_H_21_N_2_O_6_ requires 385.1400.

##### 4-(4-Dimethylaminophenyl)-1-(3,4,5-trimethoxyphenyl)-3-vinylazetidin-2-one (**7e**)

Preparation as described above from crotonyl chloride and (4-(dimethylamino)benzylidene)-3,4,5-trimethoxyphenylamine (**5e**) to afford the product as a brown oil, yield 61%. IR (NaCl) ν_max_: 1746 (C=O) cm^−1^. ^1^H NMR (400 MHz, CDCl_3_): δ 2.98 (s, 6H), 3.73 (s, 6H), 3.77 (s, 3H), 3.78–3.92 (m, 1H), 4.70 (d, *J* = 2.00 Hz, 1H), 5.30–5.40 (m, 2H), 5.99–6.08 (m, 1H), 6.60 (s, 2H), 6.74 (d, *J* = 8.04 Hz, 2H), 7.27 (d, *J* = 9.04 Hz, 2H). ^13^C NMR (100 MHz, CDCl_3_): δ 40.01, 55.55, 60.49, 61.27, 63.30, 94.27, 112.20, 119.15, 126.67, 130.38, 133.58, 134.62, 137.93, 147.60, 152.95, 165.25. HRMS: found 381.1819 (M^+^-H); C_22_H_25_N_2_O_4_ requires 381.1814.

##### 4-Phenyl-1-(3,4,5-trimethoxyphenyl)-3-vinylazetidin-2-one (**7f**)

Preparation as described above from crotonyl chloride and benzylidene-(3,4,5-trimethoxyphenyl)amine (**5f**) to afford the product as a yellow solid, yield 29%, Mp 109–111 °C. IR (KBr) ν_max_: 1750 (C=O) cm^−1^. ^1^H NMR (400 MHz, CDCl_3_): δ 3.72 (s, 6H), 3.78 (s, 3H), 3.80–3.81 (m, 1H), 4.79 (d, *J* = 2.52 Hz, 1H), 5.35–5.43 (m, 2H), 6.01–6.06 (m, 1H), 6.56 (s, 2H), 7.39–7.41 (m, 5H). ^13^C NMR (100 MHz, CDCl_3_): δ 55.54, 60.51, 61.22, 63.35, 94.22, 119.57, 125.44, 125.53, 128.31, 128.80, 130.02, 133.34, 133.97, 136.85, 153.04, 164.75. HRMS: found 338.1383 (M^+^-H); C_20_H_20_NO_4_ requires 338.1392.

##### 4-p-Tolyl-1-(3,4,5-trimethoxyphenyl)-3-vinylazetidin-2-one (**7g**)

Preparation as described above from crotonyl chloride and (4-methylbenzylidene)-(3,4,5-trimethoxyphenyl)amine (**5g**) to afford the product as a yellow solid, yield 35%, Mp 106–107 °C. IR (KBr) ν_max_: 1746 (C=O) cm^−1^. ^1^H NMR (400 MHz, CDCl_3_): δ 2.37 (s, 3H), 3.72 (s, 6H), 3.75 (m, 1H), 3.77 (s, 3H), 4.76 (d, *J* = 2.04 Hz, 1H), 5.32–5.41 (m, 2H), 5.99–6.08 (m, 1H), 6.56 (s, 2H), 7.21 (d, *J* = 8.00 Hz, 2H), 7.28 (d, *J* = 8.00 Hz, 2H). ^13^C NMR (100 MHz, CDCl_3_): δ 20.76, 55.54, 60.49, 61.07, 63.39, 94.22, 119.44, 125.48, 129.45, 130.11, 133.40, 133.78, 133.92, 138.18, 153.01, 164.87. HRMS: found 354.1706 (M^+^+H); C_21_H_24_NO_4_ requires 354.1705.

##### 4-(4-Methoxyphenyl)-1-(3,4,5-trimethoxyphenyl)-3-vinylazetidin-2-one (**7h**)

Preparation as described above from crotonyl chloride and (4-methoxybenzylidene)-3,4,5-trimethoxyphenylamine (**5h**) to afford the product as a brown oil, yield 34% [98]. IR (NaCl) ν_max_: 1747 (C=O) cm^−1^. ^1^H NMR (400 MHz, CDCl_3_): δ 3.64 (s, 6H), 3.67–3.68 (m, 1H), 3.69 (s, 3H), 3.72 (s, 3H), 4.70 (d, *J* = 2.00 Hz, 1H), 5.22–5.32 (m, 2H), 5.91–6.00 (m, 1H), 6.51 (s, 2H), 6.85 (d, *J* = 8.52 Hz, 2H), 7.26 (d, *J* = 8.52 Hz, 2H). ^13^C NMR (100 MHz, CDCl_3_): δ 55.37, 56.03, 60.97, 61.37, 63.92, 94.70, 114.61, 119.90, 127.31, 129.14, 130.58, 133.86, 133.93, 153.48, 159.90, 165.39. HRMS: found 370.1658 (M^+^+H); C_21_H_24_NO_5_ requires 370.1654.

##### 4-(4-Ethoxyphenyl)-1-(3,4,5-trimethoxyphenyl)-3-vinylazetidin-2-one (**7i**)

Preparation as described above from crotonyl chloride and (4-ethoxybenzylidene)-(3,4,5-trimethoxyphenyl)amine (**5i**) to afford a colourless solid, yield 33%, Mp 92–93 °C. [98] IR (KBr) ν_max_: 1749 (C=O) cm^−1^. ^1^H NMR (400 MHz, CDCl_3_): δ 1.43 (t, *J* = 6.84 Hz, 3H), 3.73 (s, 6H), 3.75–3.76 (m, 1H), 3.78 (s, 3H), 4.05 (q, *J* = 6.86 Hz, 2H), 4.73 (d, *J* = 2.48 Hz, 1H), 5.33–5.42 (m, 2H), 6.00–6.08 (m, 1H), 6.57 (s, 2H), 6.93 (d, *J* = 8.80 Hz, 2H), 7.31 (d, *J* = 8.80 Hz, 2H). ^13^C NMR (100 MHz, CDCl_3_): δ 14.33, 55.55, 60.50, 60.94, 63.11, 63.43, 94.22, 114.65, 119.40, 126.83, 128.49, 130.14, 133.42, 133.90, 153.01, 158.81, 164.94 (C=O). HRMS: found 384.1819 (M^+^+H); C_22_H_26_NO_5_ requires 384.1811.

##### 4-(4-Butoxyphenyl)-1-(3,4,5-trimethoxyphenyl)-3-vinylazetidin-2-one (**7j**)

Preparation as described above from crotonyl chloride and (4-butoxybenzylidene)-3,4,5-trimethoxyphenylamine (**5j**) to afford a yellow solid, yield 40%, Mp 100–102 °C. IR (KBr) ν_max_: 1749 (C=O) cm^−1^. ^1^H NMR (400 MHz, CDCl_3_): δ 0.98 (t, *J* = 7.32 Hz, 3H), 1.45–1.54 (m, 2H), 1.74–1.81 (m, 2H), 3.72 (s, 6H), 3.73-3.74 (m, 1H), 3.77 (s, 3H), 3.97 (t, *J* = 6.84 Hz, 2H), 4.73 (d, *J* = 1.96 Hz, 1H), 5.31–5.40 (m, 2H), 6.00–6.05 (m, 1H), 6.56 (s, 2H), 6.92 (d, *J* = 7.84 Hz, 2H), 7.30 (d, *J* = 8.80 Hz, 2H). ^13^C NMR (100 MHz, CDCl_3_): δ 13.39, 18.77, 30.78, 55.53, 60.48, 60.93, 63.43, 67.33, 94.22, 114.66, 119.36, 126.80, 128.40, 130.15, 133.42, 133.89, 153.00, 159.03, 164.93. HRMS: found 412.2129 (M^+^+H); C_24_H_30_NO_5_ requires 412.2124.

##### 4-(4-Phenoxyphenyl)-1-(3,4,5-trimethoxyphenyl)-3-vinylazetidin-2-one (**7k**)

Preparation as described above from crotonyl chloride and (4-phenoxylbenzylidene)-(3,4,5-trimethoxyphenyl)amine (**5k**) to afford a pale yellow solid, yield 37%, Mp 128–130 °C. IR (KBr) ν_max_: 1749 (C=O) cm^−1^. ^1^H NMR (400 MHz, CDCl_3_): δ 3.75 (s, 6H), 3.77–3.78 (m, 1H), 3.79 (s, 3H), 4.78 (d, *J* = 2.52 Hz, 1H), 5.35–5.44 (m, 2H), 6.01–6.10 (m, 1H), 6.57 (s, 2H), 7.02–7.05 (m, 4H), 7.14–7.17 (m, 1H), 7.35–7.39 (m, 4H). ^13^C NMR (100 MHz, CDCl_3_): δ 55.58, 60.52, 60.77, 63.42, 94.26, 118.73, 118.77, 119.58, 123.37, 127.03, 129.44, 130.00, 131.27, 133.29, 134.18, 153.07, 156.09, 157.40, 164.74. HRMS: found 454.1610 (M^+^+Na); C_26_H_25_NO_5_Na requires 454.1630.

##### 4-(4-Benzyloxyphenyl)-1-(3,4,5-trimethoxyphenyl)-3-vinylazetidin-2-one (**7l**)

Preparation as described above from crotonyl chloride and (4-benzyloxybenzylidene)-3,4,5-trimethoxyphenylamine (**5l**) to afford a cream solid, yield 37%, Mp 148–149 °C. IR (KBr) ν_max_: 1746 (C=O) cm^−1^. ^1^H NMR (400 MHz, CDCl_3_): δ 3.72 (s, 6H), 3.75–3.76 (m, 1H), 3.78 (s, 3H), 4.74 (d, *J* = 2.52 Hz, 1H), 5.09 (s, 2H), 5.33–5.42 (m, 2H), 5.99–6.08 (m, 1H), 6.56 (s, 2H), 7.02 (d, *J* = 8.56 Hz, 2H), 7.31–7.46 (m, 7H). ^13^C NMR (100 MHz, CDCl_3_): δ 56.03, 60.97, 61.36, 63.89, 70.08, 94.69, 115.57, 119.92, 127.36, 127.48, 128.14, 128.67, 129.44, 130.58, 133.86, 136.61, 139.50, 153.49, 159.05, 165.37. HRMS: found 468.1774 (M^+^+Na); C_27_H_27_NO_5_Na requires 468.1787.

##### 4-Naphthalen-1-yl-1-(3,4,5-trimethoxyphenyl)-3-vinylazetidin-2-one (**7m**)

Preparation as described above from crotonyl chloride and naphthalen-1-ylmethylene-(3,4,5-trimethoxyphenyl)amine (**5m**) to afford the product as a yellow solid, yield 34%, Mp 121–122 °C. IR (KBr) ν_max_: 1754 (C=O) cm^−1^. ^1^H NMR (400 MHz, CDCl_3_): δ 3.71 (s, 6H), 3.76–3.78 (m, 1H), 3.82 (s, 3H), 5.44–5.49 (m, 2H), 5.60 (d, *J* = 2.00 Hz, 1H), 6.22–6.31 (m, 1H), 6.67 (s, 2H), 7.44–8.04 (m, 7H). ^13^C NMR (100 MHz, CDCl_3_): δ 55.69, 58.42, 60.54, 63.11, 94.49, 120.72, 122.29, 125.10, 125.74, 126.28, 127.86, 128.22, 128.75, 129.96, 130.67, 132.21, 133.45, 133.62, 134.13, 153.19, 164.82. HRMS: found 390.1715 (M^+^+H); C_24_H_24_NO_4_ requires 390.1705.

##### 4-Naphthalen-2-yl-1-(3,4,5-trimethoxyphenyl)-3-vinylazetidin-2-one (**7n**)

Preparation as described above from crotonyl chloride and naphthalen-2-ylmethylene-(3,4,5-trimethoxyphenyl)amine (**5n**) to afford the product as a yellow solid, yield 30%, Mp 145–146 °C. IR (KBr) ν_max_: 1749 (C=O) cm^−1^. ^1^H NMR (400 MHz, CDCl_3_): δ 3.69 (s, 6H), 3.77 (s, 3H), 3.84–3.87 (m, 1H), 4.97 (d, *J* = 2.52 Hz, 1H), 5.37–5.45 (m, 2H), 6.06–6.12 (m, 1H), 6.62 (s, 2H), 7.47–7.92 (m, 7H). ^13^C NMR (100 MHz, CDCl_3_): δ 55.58, 60.50, 61.38, 63.44, 94.25, 119.71, 122.47, 124.97, 126.13, 126.34, 127.39, 127.42, 129.01, 130.00, 132.86, 132.94, 133.45, 134.05, 134.33, 153.08, 164.80. HRMS: found 390.1714 (M^+^+H); C_24_H_24_NO_4_ requires 390.1705.

##### 4-(4-Methoxy-3-nitrophenyl)-1-(3,4,5-trimethoxyphenyl)-3-vinylazetidin-2-one (**7p**)

Preparation as described above from crotonyl chloride and (4-methoxy-3-nitrobenzylidene)-(3,4,5-trimethoxyphenyl)amine (**5p**) to afford a brown oil, yield 14%. IR (NaCl) ν_max_: 1754 (C=O) cm^−1^. ^1^H NMR (400 MHz, CDCl_3_): δ 3.75 (s, 6H), 3.78 (s, 3H), 3.85–3.86 (m, 1H), 3.99 (s, 3H), 4.80 (d, *J* = 2.00 Hz, 1H), 5.37–5.43 (m, 2H), 5.98–6.07 (m, 1H), 6.52 (s, 2H), 7.14–7.16 (m, 1H), 7.54–7.57 (m, 1H), 7.90 (br s, 1H). ^13^C NMR (100 MHz, CDCl_3_): δ 55.61, 56.27, 59.66, 60.52, 63.43, 94.23, 114.14, 120.19, 123.18, 129.23, 129.38, 130.74, 132.80, 134.36, 138.76, 152.65, 153.23, 164.19. HRMS: found 437.1326 (M^+^+Na); C_21_H_22_N_2_O_7_Na requires 437.1325.

##### 4-[4-Oxo-1-(3,4,5-trimethoxyphenyl)-3-vinyl-azetidin-2-yl]benzonitrile (**7q**)

Preparation as described above from 4-[(3,4,5-trimethoxyphenylimino)methyl]benzonitrile (**5q**) and *trans*-crotonyl chloride to afford the product as a colourless oil (31%). IR ν_max_ 1756.1(CO), 2312.0 (CN) cm^−1^. ^1^H NMR (400 MHz, CDCl_3_): δ 3.76 (s, br, 10H), 4.86 (d, *J* = 2.52 Hz, 1H), 5.40 (m, 2H), 5.90–6.08 (m, 1H), 6.48 (s, 2H), 7.49 (d, 2H), 7.71 (d, 2H). HRMS: Found: 387.1335 (M^+^+Na); C_21_H_20_N_2_O_4_Na requires 387.1321.

##### 4-(4-Methylsulfanylphenyl)-1-(3,4,5-trimethoxyphenyl)-3-vinylazetidin-2-one (**7r**)

Preparation as described above from crotonyl chloride and *N*-(3,4,5-trimethoxybenzylidene)-4-methylsulfanylphenylamine (**5r**), yield 17%, brown oil [98]. IR (NaCl ν max): 1744 (C=O) cm^−1^. ^1^H NMR (400 MHz, CDCl_3_): δ 2.48 (s, 3H), 3.72–3.76 (m, 10H, OMe), 4.75 (d, *J* = 2.33Hz, 1H), 5.31–5.40 (m, 2H), 5.98–6.05 (m, 1H), 6.54 (s, 2H), 7.25–7.31 (m, 4H). ^13^C NMR (100 MHz, CDCl_3_): δ15.55, 55.88, 55.61, 60.77, 60.47, 63.39, 94.23, 119.59, 125.98, 126.09, 129.96, 133.25, 133.49, 139.00, 141.19, 153.05, 164.70. HRMS: found 408.1230 (M^+^+Na); C_21_H_23_NO_4_SNa, requires 408.1245.

##### 1-(3,5-Dimethoxyphenyl)-4-(4-methoxyphenyl)-3-vinylazetidin-2-one (**7u**)

Preparation as described above from imine **5s** and crotonyl chloride to afford the product as brown oil; Yield: 17%, IR ν_max_: 1748.72 cm^−1^ (C=O, β- lactam). δ ^1^H NMR (400 MHz, CDCl_3_): 3.69 (s, 7 H), 3.79 (s, 3 H), 4.68 (s, 1 H), 5.27–5.39 (m, 2 H), 5.94–6.06 (m, 1 H), 6.15 (s, 1 H), 6.48 (s, 2 H), 6.89 (d, *J* = 7.93 Hz, 2 H), 7.27 (d, *J* = 8.54 Hz, 2 H). ^13^C NMR (100 MHz, CDCl_3_): **δ** 55.31, 55.49, 61.24, 63.97, 95.66, 96.27, 114.58, 119.77, 127.14, 129.16, 130.54, 139.23, 159.79, 161.06, 165.63. HRMS: found 340.1540 (M^+^ + H); C_20_H_22_NO_4_ requires 340.1549.

##### 4-(3,4,5-Trimethoxyphenyl)-1-(4-fluorophenyl)-3-vinylazetidin-2-one (**8a**)

Preparation as described above from crotonyl chloride and *N*-(3,4,5-trimethoxybenzylidene)-4-fluorophenylamine (**6a**) as a yellow oil, yield 18%. IR (NaCl) ν_max_: 1751 (C=O) cm^−1^. ^1^H NMR (400 MHz, CDCl_3_): δ 3.77–3.79 (m, 1H), 3.83 (s, 6H), 3.86 (s, 3H), 4.71 (d, *J* = 2.48 Hz, 1H), 5.35–5.43 (m, 2H), 6.00–6.09 (m, 1H), 6.54 (s, 2H), 6.96–7.00 (m, 2H), 7.29–7.32 (m, 2H). ^13^C NMR (100 MHz, CDCl_3_): δ 55.76, 60.43, 61.39, 63.73, 101.99, 115.34, 115.57, 117.98, 118.06, 119.84, 129.87, 132.18, 133.36, 137.63, 153.54, 159.88, 164.74. HRMS: found 358.1454 (M^+^+H); C_20_H_21_FNO_4_ requires 358.1455.

##### 4-(3,4,5-Trimethoxyphenyl)-1-(4-chlorophenyl)-3-vinylazetidin-2-one (**8b**)

Preparation as described above from crotonyl chloride and *N*-(3,4,5-trimethoxybenzylidene)-4-chlorophenylamine (**6b**) as a yellow solid, yield 32%, Mp 131–133 °C. IR (KBr) ν_max_: 1753 (C=O) cm^−1^. ^1^H NMR (400 MHz, CDCl_3_): δ 3.78–3.79 (m, 1H), 3.83 (s, 6H), 3.87 (s, 3H), 4.71 (d, *J* = 1.00 Hz, 1H), 5.36–5.43 (m, 2H), 6.02–6.08 (m, 1H), 6.54 (s, 2H), 7.24 (d, *J* = 5.88 Hz, 2H), 7.28 (d, *J* = 5.88 Hz, 2H). ^13^C NMR (100 MHz, CDCl_3_): δ 56.11, 60.70, 61.67, 64.09, 102.43, 118.11, 120.10, 129.04, 130.08, 130.24, 132.33, 135.94, 138.15, 153.91, 165.18. HRMS: found 396.0966 (M^+^+Na); C_20_H_20_^35^ClNO_4_Na requires 396.0979.

##### 4-(3,4,5-Trimethoxyphenyl)-1-(4-bromophenyl)-3-vinylazetidin-2-one (**8c**)

Preparation as described above from crotonyl chloride and *N*-(3,4,5-trimethoxy benzylidene)-4-bromophenylamine (**6c**) as a colourless solid, yield 32%, Mp 120–122 °C. IR (KBr) ν_max_: 1754 (C=O) cm^−1^. ^1^H NMR (400 MHz, CDCl_3_): δ 3.77–3.79 (m, 1H), 3.83 (s, 6H), 3.86 (s, 3H), 4.71 (d, *J* = 2.48 Hz, 1H), 5.36–5.44 (m, 2H), 6.00–6.07 (m, 1H), 6.53 (s, 2H), 7.22 (d, *J* = 8.56 Hz, 2H), 7.40 (d, *J* = 8.52 Hz, 2H). ^13^C NMR (100 MHz, CDCl_3_): δ 55.79, 60.43, 61.33, 63.82, 101.97, 116.38, 118.16, 119.93, 129.72, 131.67, 132.00, 136.05, 137.70, 153.58, 164.96. HRMS: found 440.0466 (M^+^+Na); C_20_H_20_^79^BrNO_4_Na requires 440.0473.

##### 4-(3,4,5-Trimethoxyphenyl)-1-(4-nitrophenyl)-3-vinylazetidin-2-one (**8d**)

Preparation as described above from crotonyl chloride and *N*-(3,4,5-trimethoxybenzylidene)-4-nitrophenylamine (**6d**) as a yellow oil, yield 42%. IR (NaCl) ν_max_: 1762 (C=O) cm^−1^. ^1^H NMR (400 MHz, CDCl_3_): δ 3.85 (s, 6H), 3.87 (s, 3H), 3.89–3.91 (m, 1H), 4.81 (d, *J* = 2.52 Hz, 1H), 5.40–5.47 (m, 2H), 6.01–6.10 (m, 1H), 6.54 (s, 2H), 7.45 (d, *J* = 9.04 Hz, 2H), 8.19 (d, *J* = 9.04 Hz, 2H). ^13^C NMR (100 MHz, CDCl_3_): δ 55.83, 60.45, 61.77, 64.16, 101.95, 116.45, 120.38, 124.85, 129.10, 131.32, 137.97, 142.11, 143.00, 153.73, 165.55. HRMS: found 383.1234 (M^+^-H); C_20_H_19_N_2_O_6_ requires 383.1243.

##### 4-(3,4,5-Trimethoxyphenyl)-1-(4-tert-butylphenyl)-3-vinylazetidin-2-one (**8e**)

Preparation as described above from crotonyl chloride and *N*-(3,4,5-trimethoxybenzylidene)-4-*tert*-butylphenylamine (**6e**) as a yellow solid, yield 22%, Mp 172–174 °C. IR (KBr) ν_max_: 1746 (C=O) cm^−1^. ^1^H NMR (400 MHz, CDCl_3_): δ 1.29 (s, 9H), 3.76–3.78 (m, 1H), 3.84 (s, 6H), 3.87 (s, 3H), 4.69 (d, *J* = 2.48 Hz, 1H), 5.33–5.42 (m, 2H), 6.98–6.07 (m, 1H), 6.58 (s, 2H), 7.27 (d, *J* = 9.04 Hz, 2H), 7.31 (d, *J* = 9.04 Hz, 2H). ^13^C NMR (100 MHz, CDCl_3_): δ 30.86, 33.96, 55.78, 60.43, 61.25, 63.56, 102.13, 116.27, 119.58, 125.45, 130.15, 132.77, 134.69, 137.51, 146.63, 153.45, 164.84. HRMS: found 396.2182 (M^+^+H); C_24_H_30_NO_4_ requires 396.2175.

##### 4-(3,4,5-Trimethoxyphenyl)-1-phenyl-3-vinylazetidin-2-one (**8f**)

Preparation as described above from crotonyl chloride and *N*-(3,4,5-trimethoxybenzylidene)phenylamine (**6f**) as a colourless solid, yield 34%, Mp 150–151 °C. IR (KBr) ν_max_: 1752 (C=O) cm^−1^. ^1^H NMR (400 MHz, CDCl_3_): δ 3.77–3.79 (m, 1H), 3.83 (s, 6H), 3.86 (s, 3H), 4.74 (d, *J* = 2.48 Hz, 1H), 5.35–5.44 (m, 2H), 6.01–6.10 (m, 1H), 6.56 (s, 2H), 7.07–7.11 (m, 1H), 7.27–7.33 (m, 4H). ^13^C NMR (100 MHz, CDCl_3_): δ 55.76, 60.43, 61.19, 63.57, 102.01, 116.60, 119.72, 123.67, 128.65, 130.02, 132.54, 137.12, 137.53, 153.49, 165.02. HRMS: found 362.1371 (M^+^+Na); C_20_H_21_NO_4_Na requires 362.1368.

##### 4-(3,4,5-Trimethoxyphenyl)-1-p-tolyl-3-vinylazetidin-2-one (**8g**)

Preparation as described above from crotonyl chloride and *N*-(3,4,5-trimethoxybenzylidene)-4-methylphenylamine (**6g**) as a yellow oil, yield 16%. IR (NaCl) ν_max_: 1749 (C=O) cm^−1^. ^1^H NMR (400 MHz, CDCl_3_): δ 2.30 (s, 3H), 3.75–3.77 (m, 1H), 3.83 (s, 6H), 3.86 (s, 3H), 4.71 (d, *J* = 2.48 Hz, 1H), 5.34–5.43 (m, 2H), 6.00–6.09 (m, 1H), 6.55 (s, 2H), 7.09 (d, *J* = 8.04 Hz, 2H), 7.23 (d, *J* = 8.56 Hz, 2H). ^13^C NMR (100 MHz, CDCl_3_): δ 20.48, 55.75, 60.42, 61.16, 63.54, 102.03, 116.54, 119.62, 129.12, 130.15, 132.66, 133.29, 134.69, 137.48, 153.45, 164.76. HRMS: found 376.1534 (M^+^+Na); C_21_H_23_NO_4_Na requires 376.1525.

##### 4-(3,4,5-Trimethoxyphenyl)-1-(4-ethylphenyl)-3-vinylazetidin-2-one (**8h**)

Preparation as described above from crotonyl chloride and *N*-(3,4,5-trimethoxybenzylidene)-4-ethylphenylamine (**6h**) as yellow crystals, yield 15%, Mp 110–112 °C. IR (KBr) ν_max_: 1749 (C=O) cm^−1^. ^1^H NMR (400 MHz, CDCl_3_): δ 1.21 (t, *J* = 7.48 Hz, 3H), 2.57–2.63 (q, *J* = 7.52 Hz, 2H), 3.75–3.78 (m, 1H), 3.84 (s, 6H), 3.87 (s, 3H), 4.71 (d, *J* = 2.52 Hz, 1H), 5.34–5.43 (m, 2H), 6.00–6.09 (m, 1H), 6.57 (s, 2H), 7.12 (d, *J* = 8.76 Hz, 2H), 7.25–7.29 (m, 2H). ^13^C NMR (100 MHz, CDCl_3_): δ 15.62, 28.34, 56.23, 60.89, 61.67, 64.00, 102.54, 117.07, 120.07, 128.42, 130.63, 133.18, 135.37, 137.98, 140.20, 153.93, 165.25. HRMS: found 390.1666 (M^+^+Na). C_22_H_25_NO_4_Na requires 390.1681.

##### 4-(3,4,5-Trimethoxyphenyl)-1-(4-methoxyphenyl)-3-vinylazetidin-2-one (**8i**)

Preparation as described above from crotonyl chloride and *N*-(3,4,5-trimethoxybenzylidene)-4-methoxyphenylamine (**6i**) as a yellow oil, yield 18%. IR (NaCl) ν_max_: 1744 (C=O) cm^−1^. ^1^H NMR (400 MHz, CDCl_3_): δ 3.78 (s, 3H), 3.80–3.82 (m, 1H), 3.83 (s, 6H), 3.86 (s, 3H), 4.69 (d, *J* = 2.48 Hz, 1H), 5.34–5.43 (m, 2H), 6.03–6.05 (m, 1H), 6.55 (s, 2H), 6.83 (d, *J* = 9.00 Hz, 2H), 7.28 (d, *J* = 9.00 Hz, 2H). ^13^C NMR (100 MHz, CDCl_3_): δ 54.99, 55.75, 60.42, 61.30, 63.59, 102.06, 113.85, 117.92, 119.62, 130.20, 130.68, 132.62, 137.49, 153.46, 155.67, 164.44. HRMS: found 392.1479 (M^+^+Na); C_21_H_23_NO_5_Na requires 392.1474.

##### *N*-(4-(2-(3,4,5-Trimethoxyphenyl)-4-oxo-3-vinylazetidin-1-yl)phenyl)acetamide (**8j**)

Preparation as described above from crotonyl chloride and *N*-((*E*)-4-(3,4,5-trimethoxybenzylideneamino)phenyl)acetamide (**6j**) as a colourless solid, yield 14%, Mp 223–224 °C. IR (KBr) ν_max_: 1742 (C=O) 1689 (C=O) cm^−1^. ^1^H NMR (400 MHz, CDCl_3_): δ 2.16 (s, 3H), 3.76–3.78 (m, 1H), 3.82 (s, 6H), 3.86 (s, 3H), 4.71 (d, *J* = 2.52 Hz, 1H), 5.34–5.42 (m, 2H), 6.00–6.09 (m, 1H), 6.54 (s, 2H), 7.27 (d, J = 8.56 Hz, 2H), 7.43 (d, *J* = 8.56 Hz, 2H), 7.49 (s, 1H). ^13^C NMR (100 MHz, CDCl_3_): δ 23.99, 55.75, 60.43, 61.26, 63.63, 102.04, 117.16, 119.80, 120.27, 129.99, 132.36, 133.41, 133.70, 137.52, 153.49, 164.76, 167.92. HRMS: found 419.1579 (M^+^+Na); C_22_H_24_N_2_O_5_Na requires 419.1583.

##### 1-(4-(Methylthio)phenyl)-4-(3,4,5-trimethoxyphenyl)-3-vinylazetidin-2-one (**8k**)

Preparation as described above from crotonyl chloride and *N*-(3,4,5-trimethoxybenzylidene)-4-thiomethylphenylamine (**6k**) as a colourless solid, yield 41%, Mp 104–106 °C. IR (NaCl, film) ν_max_: 1748.72 cm^−1^ (C=O, β-lactam). ^1^H NMR (400 MHz, CDCl_3_): δ ppm 2.41 (s, 3 H), 3.73 (d, *J* = 7.93, 1.22 Hz, 1 H), 3.77–3.85 (m, 9 H), 4.67 (d, *J* = 2.44 Hz, 1 H), 5.30–5.40 (m, 2 H), 5.98 (d, *J* = 10.07, 7.63 Hz, 1 H), 6.50 (s, 2 H), 7.13–7.18 (m, 2 H), 7.23 (d, *J* = 7.32 Hz, 2 H). ^13^C NMR (100 MHz, CDCl_3_): δ ppm 16.50, 56.21, 60.83, 61.70, 64.07, 102.52, 117.59, 120.12, 127.90, 130.40, 132.77, 133.52, 135.16, 153.95, 165.19. HRMS: found 408.1255 (M^+^+Na); C_21_H_23_NNaO_4_S requires 408.1246.

#### 4.1.4. 4-[3-Hydroxy-4-methoxyphenyl]-1-(3,4,5-trimethoxyphenyl)-3-vinylazetidin-2-one (**7s**)

To a stirring, refluxing solution of the TBDMS protected imine **5o** (5 mmol) and triethylamine (6 mmol) in anhydrous dichloromethane (40 mL), a solution of crotonyl chloride (6 mmol) in anhydrous dichloromethane (10 mL) was added over 45 min under nitrogen. The reaction was kept at reflux for 5 h and then at room temperature overnight (16 h), until the starting material had disappeared as monitored by TLC in (1:1 *n*-hexane: ethyl acetate). The reaction mixture was washed with water (2 × 100 mL). The combined organic extract was dried over anhydrous Na_2_SO_4_ before the solvent was removed under reduced pressure. The crude product was purified by flash chromatography over silica gel (eluant: *n*-hexane: ethyl acetate, 4:1) to afford the β-lactam **7o** as an oil. To a stirring solution of the protected β-lactam **7o** (5 mmol) under N_2_ and 0 °C in dry THF was added dropwise 1.5 equivalents of 1.0 M *tert*-butylammonium fluoride (*t*-BAF) solution in hexanes (5 mmol). The resulting solution was left to stir at 0 °C until reaction was complete as monitored by TLC. The reaction mixture was diluted with ethyl acetate (75 mL) and washed with 0.1M HCl (100 mL). The aqueous layer was further extracted with ethyl acetate (2 × 25 mL). All organic layers were combined and washed with water (100 mL) and saturated brine (100 mL) before being dried over anhydrous sodium sulphate. The solvent was removed under reduced pressure to yield the phenol which was further purified by flash chromatography over silica gel (eluent: *n*-hexane: ethyl acetate, 4:1) to afford the product as a yellow oil, yield 20%. IR (NaCl, film) ν_max_: 3367 (OH), 1749 (C=O, β-lactam), 1587, 1501, 1235, 1127 cm^−1^. ^1^H NMR (400 MHz, CDCl_3_): δ 3.74 (s, 6H), 3.74–3.75 (m, 1H), 3.78 (s, 3H), 3.91 (s, 3H), 4.69 (d, *J* = 2.52 Hz, 1H), 5.32–5.40 (m, 2H), 5.77 (br s, 1H), 5.98–6.05 (m, 1H), 6.57 (s, 2H), 6.87–6.96 (m, 3H). ^13^C NMR (100 MHz, CDCl_3_): δ 55.56, 55.58, 60.50, 60.86, 63.38, 94.24, 110.50, 111.54, 117.30, 119.44, 129.90, 130.08, 133.38, 133.91, 145.82, 146.36, 153.01, 164.86. HRMS: found 408.1434 (M^+^+Na); C_21_H_23_NO_6_Na requires 408.1423.

#### 4.1.5. 4-(3-Amino-4-methoxyphenyl)-1-(3,4,5-trimethoxyphenyl)-3-vinylazetidin-2-one (**7t**)

To a flask containing the 4-(4-methoxy-3-nitrophenyl)-1-(3,4,5-trimethoxyphenyl)-3-vinylazetidin-2-one (**7p**) (0.25 mmol) and zinc powder 10 μm (2.5 mmol) was added 15 mL of acetic acid at room temperature under N_2_ and reaction left to stir for 7 days. The reaction was filtered through a celite pad and the filtrate collected. Solvent was removed under reduced pressure and purified by flash chromatography over silica gel (elutent: ethyl acetate: *n*-hexane, 1:1) to yield the title compound as a brown solid, yield 43%, Mp 100–101 °C. IR (KBr) ν_max_: 3370 (NH_2_), 1747 (C=O) cm^−1^. ^1^H NMR (400 MHz, CDCl_3_): δ 3.75 (s, 6H), 3.79 (s, 3H), 3.87–3.88 (m, 1H), 3.88 (s, 3H), 4.65 (d, *J* = 2.52 Hz, 1H), 5.31–5.41 (m, 2H), 5.98–6.07 (m, 1H), 6.60 (s, 2H), 6.72–6.78 (m, 3H). ^13^C NMR (100 MHz, CDCl_3_): δ 55.11, 55.58, 60.50, 61.15, 63.35, 94.23, 109.93, 111.06, 115.86, 119.29, 129.26, 130.25, 133.53, 134.01, 136.50, 147.09, 152.99, 165.07. HRMS: found 407.1597(M^+^+Na); C_21_H_24_N_2_O_5_Na requires 407.1583.

#### 4.1.6. 4-(4-Methoxyphenyl)-1-(3,4,5-trimethoxyphenyl)azetidin-2-one (**9a**)

Zinc powder (9 mmol) was activated using trimethylchlorosilane (0.5 mmol) in anhydrous benzene (1 mL) by heating for 15 min at 40 °C and followed by 5 min at 100 °C in a microwave. After cooling, the imine **5h** (2 mmol) and ethyl bromoacetate (2.4 mmol) were added to the reaction vessel and the mixture was placed in the microwave for 30 min at 100 °C. The reaction mixture was filtered through Celite to remove the zinc catalyst and then diluted with dichloromethane. This solution was washed with saturated ammonium chloride solution (20 mL) and 25% ammonium hydroxide (20 mL) and then with dilute HCl (40 mL), followed by water (40 mL). The organic phase was dried over anhydrous sodium sulphate and the solvent was removed under reduced pressure. The crude product was purified by flash column chromatography over silica gel (eluent: *n*-hexane: ethyl acetate, 2:1) to afford the product as a yellow solid, 39%, 267 mg, Mp 60–62 °C [87]. Purity (HPLC): 99.6%. IR (KBr) ν_max_: 2938, 1747 (C=O), 1603, 1507, 1246, 1126 cm^−1^. ^1^H NMR (400 MHz, CDCl_3_): δ 2.96 (dd, *J* = 15.20, 2.50 Hz, 1H), 3.55 (dd, *J* = 15.18, 5.86 Hz, 1H), 3.73 (s, 6H), 3.77 (s, 3H), 3.83 (s, 3H), 4.94 (dd, *J* = 5.84, 2.36 Hz, 1H), 6.56 (s, 2H), 6.93 (d, *J* = 8.76 Hz, 2H), 7.34 (d, *J* = 8.76 Hz). ^13^C NMR (100 MHz, CDCl_3_): δ 46.49, 53.66, 54.88, 55.54, 60.49, 93.98, 114.09, 126.84, 129.53, 133.62, 133.93, 152.99, 159.34, 164.16. HRMS: found 344.1506 (M^+^+H); C_19_H_22_NO_5_ requires 344.1498.

#### 4.1.7. 4-(3-Hydroxy-4-methoxyphenyl)-1-(3,4,5-trimethoxyphenyl)-azetidin-2-one (**9c**)

(i) Zinc powder (458 mg, 7 mmol (method A) or 21 mmol (method B)) and chlorotrimethylsilane (0.32 mL, 2.5 mmol) were refluxed for 3 min in anhydrous benzene (10 mL) under N_2_ and then allowed to cool. To the cooled stirring solution, the appropriately substituted imine (**5o**) (5 mmol) and ethylbromoacetate (0.66 mL, 6 mmol) were added and refluxed for 7 h. The reaction was cooled to 0 °C and poured onto NH_4_Cl (sat), (10 mL) and 30% NH_4_OH (10 mL). The resulting solution was extracted with DCM (2 × 20 mL) and the organic layer further washed with 0.1N HCl (20 mL) and water (20 mL) before being dried over Na_2_SO_4_, filtered and the solvent removed under reduced pressure to afford the protected product (**9b**), yield 37%, 876 mg (method A), 77%, 1.823 g (method B) as a pale brown resin which was used immediately in the following reaction. IR (NaCl ٧ max): 1749 (C=O) cm^−1^. ^1^H NMR (400 MHz, CDCl_3_): δ 0.07 (s, 3H, OTBDMS), δ0.09 (s, 3H), 2.91–2.95 (dd, *J* = 2.47 Hz, 15.04, 1H), 3.48–3.53 (dd, *J* = 5.52 Hz, 15.55 Hz, 1H), 3.71 (s, 6H), 3.76 (s, 3H), 3.80 (s, 3H), 4.86–4.88 (dd, *J* = 2.52 Hz, 5.52, 1H), 6.54 (s, 2H), 6.83–6.94 (m, 3H). ^13^C NMR (100 MHz, CDCl_3_): –5.19, –5.17, 17.98, 25.02, 46.41, 53.59, 55.03, 60.46, 55.49, 94.04, 111.82, 117.99, 119.05, 129.98, 133.57, 133.74, 145.15, 150.74, 152.95, 164.14. (ii) To a stirring solution of the silyl ether β-lactam (**9b**) (4 mmol) in dry THF (30 mL) was added a solution of 1.0M tBAF in hexanes (4 mL, 4 mmol) under N_2_ at 0 °C. The reaction mixture was stirred for a further 90 min. Reaction was diluted with ethyl acetate (150 mL) and washed with 0.1M HCl (200 mL). The aqueous layer was further extracted with ethyl acetate (2 × 50 mL). All the organic layers were collected and washed with water (200 mL) and saturated brine (200 mL) before being dried over Na_2_SO_4_. Solvent was removed under reduced pressure and the phenol was isolated by flash chromatography over silica gel (eluent: *n*-hexane: ethyl actetate, 1:1) to afford the desired product, yield 73%, 1.05 g, as a yellow solid, Mp 78–80 °C [87]. IR (NaCl ٧ max): 1741 cm^−1^, 3443 cm^−1^, 2937 cm^−1^. ^1^H NMR (400 MHz, CDCl_3_): δ2.88–2.93 (dd, *J* = 2.48 Hz, 15.06 Hz, 1H), 3.45–3.50 (dd, *J* = 5.52 Hz, 15.56Hz, 1H), 3.67 (s, 6H), 3.72 (s, 3H), 3.84 (s, 3H), 4.84–4.86 (dd, *J* = 2.52 Hz, 5.52 Hz, 1H), 6.14 (s, 1H), 6.53 (s, 2H), 6.81–6.93 (m, 3H). ^13^C NMR (100 MHz, CDCl_3_): δ 46.27, 53.68, 60.44, 55.52, 94.04, 110.57, 111.63, 117.36, 130.61, 133.56, 133.75, 145.90, 146.48, 152.94, 164.27. HRMS: Found 382.1251 (M^+^+Na); C_19_H_21_NO_6_Na requires 382.1267.

#### 4.1.8. 3-(1-Hydroxyethyl)-4-(3-hydroxy-4-methoxyphenyl)-1-(3,4,5-trimethoxyphenyl) azetidin-2-one (**10a**)

To a solution of the TBDMS protected 3-unsubstituted β-lactam (**9b**) (125 mg, 0.264 mmol) in dry THF (3 mL) under N_2_ at −78 °C (dry ice and acetone) was added 2.0 M LDA solution (0.264 mL, 0.528 mmol). The resulting solution was left to stir for 5 min before a solution of acetaldehyde (49 mg, 0.396 mmol) in dry THF (1.5 mL) was added. The reaction was left to stir for 30 min at −78 °C, then poured onto saturated NaCl solution (25 mL). The resulting solution was extracted with ethyl acetate (50 mL) and the solvent was dried over Na_2_SO_4_ before being removed under reduced pressure. Preliminary purification was achieved by passage through a short pad (5 cm) of silica (eluent: DCM) to yield the OTBDMS protected ether **10i** as an oil. To a stirring solution of the OTBDMS protected ether **10i** (2 mmol) in dry THF (10 mL) was added a solution of 1.0 M TBAF in hexanes (2 mL, 2 mmol) under N_2_ at 0 °C. The reaction mixture was stirred for a further 90 min then diluted with ethyl acetate (75 mL) and washed with 0.1 M HCl (100 mL). The aqueous layer was further extracted with ethyl acetate (2 × 25 mL). All the organic extracts were collected and washed with H_2_O (100 mL), and saturated brine (100 mL) before being dried over Na_2_SO_4_ and solvent was removed under reduced pressure. Purification was carried out by chromatography using a Biotage SP1 chromatography system using a +12M column and detection set at 280 nM and a fraction volume of 12 mL. A gradient elution of 2% ethyl acetate in *n*-hexane to 100% ethyl acetate over 15 column volumes was used. The desired product was obtained as a brown oil, 36 mg, yield 17% [99] IR (NaCl ٧ max): 1738 (C=O), 3427 (OH) cm^−1^. ^1^H NMR (400 MHz, CDCl_3_): δ1.33 (d, *J* = 6.28 Hz, 1H), 1.40 (d, *J* = 6.52 Hz, 2H), 2.58 (br s, 1H), 3.14 (m, 1H), 3.72 (s, 6H), 3.76 (s, 3H), 3.89 (s, 3H), 4.24 (q, *J* = 6.04 Hz, 0.66H), 4.36 (q, *J* = 5.76 Hz, 0.33H), 4.77 (d, *J* = 2.28 Hz, 0.6H), 4.99 (d, *J* = 2.28, 0.4H), 5.95 (s, 0.6H), 5.96 (s, 0.4H), 6.54 (s, 2H), 6.83–7.01 (m, 3H). ^13^C NMR (100 MHz, CDCl_3_): δ 21.34, 21.52, 55.99, 56.06, 57.64 57.68, 56.68, 60.93, 64.94, 66.05, 66.10, 94.70, 94.75, 111.05, 112.16, 112.22, 117.88, 130.50, 130.97, 133.67, 134.32, 146.25, 146.32, 146.88, 153.43, δ165.89, 166.06. HRMS: Found 426.1540 (M^+^+Na); C_21_H_25_NO_7_Na requires 426.1529.

#### 4.1.9. 3-((E)-1-Hydroxybut-2-enyl)-1-(3,4,5-trimethoxyphenyl)-4-(4-methoxyphenyl) azetidin-2-one (**10b**)

Following the procedure described above for compound **10a**, using the β-lactam **9a** and crotonaldehyde, the product was obtained as a colourless solid, 82 mg, yield 25%, Mp 143–144 °C. IR (KBr) ν_max_: 3455 (OH), 1745 (C=O), 1591, 1502, 1248, 1127 cm^−1^. ^1^H NMR (400 MHz, CDCl_3_): δ 1.73–1.77 (m, 2H), 3.25–3.28 (m, 1H), 3.73 (s, 6H), 3.77 (s, 3H), 3.83 (s, 3H), 4.69–4.72 (m, 1H), 4.83 (d, *J* = 2.52 Hz, 1H), 5.57–5.62 (m, 1H), 5.80–5.88 (m, 1H), 6.56 (s, 2H), 6.91–6.94 (m, 2H), 7.28–7.32 (m, 2H). ^13^C NMR (100 MHz, CDCl_3_): δ 17.28, 54.86, 55.52, 55.90, 60.49, 64.70, 68.68, 94.25, 114.06, 114.11, 126.96, 127.81, 129.27, 129.37, 129.75, 133.31, 152.97, 159.16, 165.00. HRMS: found 436.1740 (M^+^+Na); C_23_H_27_NO_6_Na requires 436.1736.

#### 4.1.10. 3-(1-Hydroxybut-2-enyl)-4-(3-hydroxy-4-methoxyphenyl)-1-(3,4,5-trimethoxyphenyl) azetidin-2-one (**10c**)

Following the procedure described above for compound **10a**, using the β-lactam (**9b**) and crotonaldehyde, the title compound was obtained as a brown oil, 73 mg, yield 32%. IR (NaCl ٧ max): 1732 (C=O), 3427 (OH) cm^−1^. ^1^H NMR (400 MHz, CDCl_3_): δ1.72 (d, *J* = 7.0 Hz, 1.8H), 1.74 (d, *J* = 1.0Hz, 1.2H), 2.50 (br, s, 1 H), 3.24 (m, 1H), 3.73–3.79 (overlapping singlets, 9H), 3.89 (s, 3H), 4.51 (t, *J* = 6.53Hz, 0.4H), 4.67 (t, *J* = 6.80 Hz, 0.6H), 4.76 (d, *J* = 2.48 Hz, 0.4H), 4.96 (d, *J* = 2.0 Hz, 0.6H), 5.78–5.85 (m, 2H). ^13^C NMR (100 MHz, CDCl_3_): δ17.25, 55.53, 55.84, 64.38, 60.47, 64.65, 68.42, 70.60, 94.26, 110.50, 111.75, 111.82, 117.42, 117.48, 127.62, 129.22, 129.78, 130.47, 133.25, 133.27, 146.17, 146.32, 152.93, 165.21. HRMS: Found 452.1707 (M^+^+Na); C_23_H_27_NO_7_Na requires 452.1685.

#### 4.1.11. 4-(3-Hydroxy-4-methoxyphenyl)-3-(1-hydroxy-1-methylethyl)-1-(3,4,5-trimethoxy-phenyl)azetidin-2-one (**10d**)

Following the procedure described above for compound **10a**, using the β-lactam (**9b**) and acetone, the title compound was obtained as a brown oil, 51 mg, yield 23%. IR (NaCl ν max): 1732 (C=O), 3429 (OH) cm^−1^. ^1^H NMR (400 MHz, CDCl_3_): δ1.35 (s, 3H), 1.48 (s, 3H), 1.91 (br, s, 1H), 3.13 (d, *J* = 2.52 Hz, 1H), 3.72 (s, 6H), 3.76 (s, 3H), 3.89 (s, 3H), 4.89 (d, *J* = 2.52 Hz, 1H), 5.90 (br, s, 1H), 6.56 (s, 2H), 6.83–6.96 (m, 3H). ^13^C NMR (100 MHz, CDCl_3_): δ 27.36, 27.62, 55.50, 55.53, 56.73, 60.47, 69.50, 94.27, 110.57, 111.78, 117.48, 130.51, 133.18, 133.79, 145.80, 146.25, 152.93, 165.29. HRMS: found 440.1671 (M^+^+Na); C_22_H_27_NO_7_Na requires 440.1685.

#### 4.1.12. 3-((E)-1-Hydroxy-3-phenylallyl)-1-(3,4,5-trimethoxyphenyl)-4-(4-methoxyphenyl) azetidin-2-one (**10e**)

Following the procedure described above for compound **10a**, using the β-lactam **9a** and cinnamaldehyde, the product was obtained as a brown oil, 88 mg, yield 23%. IR (NaCl, film) ν_max_: 3456 (OH), 1746 (C=O), 1579, 1503, 1266, 1123 cm^−1^. ^1^H NMR (400 MHz, CDCl_3_): δ 3.38–3.39 (m, 1H), 3.73 (s, 6H), 3.78 (s, 3H), 3.82 (s, 3H), 4.53–4.56 (m, 1H), 4.78 (br s, 1H), 6.26–6.33 (m, 1H), 6.57 (s, 2H), 6.71–6.80 (m, 1H), 6.89–7.35 (m, 9H). HRMS: found 498.1885 (M^+^+Na); C_28_H_29_NO_6_Na requires 498.1893.

#### 4.1.13. 4-(3-Hydroxy-4-methoxy-phenyl)-3-(1-hydroxy-3-phenyl-allyl)-1-(3,4,5-trimethoxy-phenyl)-azetidin-2-one (**10f**)

Following the procedure described above for compound **10a**, using 3-unsubstituted β-lactam (**9b**) and cinnamaldehyde, the title compound was obtained as a brown oil, 99 mg, yield 38%. IR (NaCl ν max): 1732 (C=O) 3418 (OH) cm^−1^. ^1^H NMR (400 MHz, CDCl_3_): δ3.21 (br, s, 1H), 3.34 (dd, *J* = 2.55 Hz, 5.63 Hz, 0.53H), 3.37 (dd, *J* = 2.55 Hz, 5.62 Hz, 0.57H), 3.69 (s, 6H), 3.76 (s, 3H), 3.85 (s, 3H), 4.75 (dd, *J* = 6.13 Hz, 12.40 Hz, 0.43H), 4.86 (d, *J* = 2.45 Hz, 0.43H), 4.91 (dd, *J* = 3.82 Hz, 7.63 Hz, 0.57H), 5.03 (d, *J* = 2.45 Hz, 0.53H), 5.81–5.98 (m, 1H), 6.24 (dd, *J* = 5.54 Hz, 16.28 Hz, 0.57H), 6.41 (dd, *J* = 5.54 Hz, 16.28 Hz, 0.43H), 6.58 (s, 2H), 6.72–6.96 (m, 3H), 7.35–7.39 (m, 5H). ^13^C NMR (100 MHz, CDCl_3_): 53.32, 55.84, 56.17, 60.75, 57.35, 57.41, 64.81, 68.41, 70.77, 94.81, 94.86, 95.12, 110.61, 110.95, 110.98, 117.84, 117.91, 126.37, 126.61, 128.03, 128.46, 129.49, 130.33, 131.45, 133.12, 134.48, 136.20, 146.16, 146.61, 148.29, 153.29, 153.31, 165.48, 165.48. HRMS: found 514.1826 (M^+^+Na); C_28_H_29_NO_7_Na requires 514.1842.

#### 4.1.14. 4-(3-Hydroxy-4-methoxyphenyl)-3-(1-hydroxyallyl)-1-(3,4,5-trimethoxy phenyl) azetidin-2-one (**10g**)

Following the procedure described above for compound **10a**, using 3-unsubstituted β-lactam (**9b**) and acrolein, the title compound was obtained as a brown oil, 48 mg, yield 22%. IR (NaCl ν max): 1732 (C=O), 3428 (OH) cm^−1^. ^1^H NMR (400 MHz, CDCl_3_): δ 1.78 (br, s, 1H), 2.42 (br, s, 1H), 3.24–3.30 (m, 1H), 3.73 (s, 6H), 3.77 (s, 3H), 3.91 (s, 3H), 6.60 (t, *J* = 6.4 Hz, 6.6 Hz, 0.58H), 4.74 (dd, *J* = 6.4 Hz, 6.5 Hz, 0.32H), 4.80 (d, *J* = 2.3 Hz, 0.58H), 4.96 (d, *J* = 2.7 Hz, 0.32H), 5.28–5.44 (m, 2H), 5.78 (br, s, 1H), 5.90–5.98 (m, 0.32H), 6.02–6.10 (m, 0.68H), 6.56 (s, 2H), 6.84–7.28 (m, 3H). ^13^C NMR (100 MHz, CDCl_3_): δ 55.53, 55.55, 55.62, 60.48, 56.85, 63.99, 68.28, 70.76, 94.34, 110.47, 110.52, 111.74 111.81, 115.56, 117.47, 117.51, 129.93, 130.20, 133.20, 136.59, 136.80, 145.79, 146.14, 146.33, 152.96, 164.89. HRMS: found 438.1547 (M^+^+Na); C_22_H_25_NO_7_Na requires 438.1529.

#### 4.1.15. 4-(3-Hydroxy-4-methoxy-phenyl)-3-(1-hydroxy-1-methylallyl)-1-(3,4,5-trimethoxy-phenyl)-azetidin-2-one (**10h**)

Following the procedure described above for compound **10a**, using 3-unsubstituted β-lactam (**9b**) and 3-buten-2-one, the title compound was obtained as a brown oil, 41mg, yield 18%. IR (NaCl ν max): 1734 (C=O), 3433 (OH) cm^−1^. ^1^H NMR (400 MHz, CDCl_3_): δ1.46 (s, 1.53H), 1.56 (s, 1.34H), 1.8 (br, s, 1H), 3.17 (d, *J* = 2.40 Hz, 0.45H), 3.25 (d, *J* = 2.42 Hz, 0.42H), 3.71 (s, 6H), 3.76 (s, 3H), 3.88 (s, 3H), 4.81 (d, *J* = 2.40 Hz, 0.47H), 4.86 (d, *J* = 2.41 Hz, 0.44H), 5.17–5.23 (m, 1H), 5.31–5.42 (m, 1H), 5.81–6.81 (m, 1H), 5.81–5.83 (overlapping singlets, 1H), 6.54 (s, 2H), 6.83–6.96 (m, 3H). ^13^C NMR (100 MHz, CDCl_3_): δ 26.06, 26.19, 53.00, 56.01, 60.46, 55.51, 55.54, 67.47, 68.46, 71.70, 71.76, 94.28, 110.46, 110.54, 111.84, 111.92, 117.51, 117.60, 130.35, 133.17, 133.81, 140.73, 141.39, 141.46, 145.66, 146.13, 152.94, 164.45, δ164.91. HRMS: found 452.1691 (M^+^Na); C_23_H_27_NO_7_Na requires 452.1685.

#### 4.1.16. 3-Acetyl-4-(3-hydroxy-4-methoxyphenyl)-1-(3,4,5-trimethoxyphenyl)azetidin-2-one (**11**)

**Method A:** To a stirring solution of pyridinium chlorochromate (132 mg, 0.57 mmol) in dry DCM (2 mL) under N_2_ at room temperature was added quickly a solution of the silyl protected β-lactam (**10i**) (195 mg, 0.38 mmol). The reaction was stirred at room temperature for 18 h and then diluted with diethyl ether (25 mL) and the resulting suspension allowed to settle and the diethyl ether layer decanted off. The remaining solid was washed and decanted twice with two further 25 mL portions of diethylether. The organic extracts were combined and dried over MgSO_4_, filtered, and the solvent removed under reduced pressure. The *t*BDMS ether was removed by treatment with tBAF as previously described, to afford the title compound as an oil, 7%, 11mg. **Method B:** The 3-unsubstituted β-lactam (**9b**) (378 mg, 0.80 mmol) was dissolved in THF (7 mL) in a dry flask flushed with N_2_ and cooled to −78 °C. To this stirring solution LDA (1.0 M solution, 0.8 mL, 0.8 mmol) was added all at once and the reaction left to stir for 5 min prior to the dropwise addition of acetyl chloride (0.085 mL, 1.2 mmol), in THF (2 mL). The reaction mixture was allowed to stir at −78 °C for 30 min then stirred at room temperature for 5 min before being poured into saturated brine (50 mL). The brine solution was extracted with ethyl acetate (2 × 50 mL), the organic layers combined, dried over MgSO_4_, filtered, and the solvent removed under reduced pressure. Purification by flash column chromatography over silica gel (eluent: *n*-hexane: ethyl acetate, 1:1) followed by removal of the TBDMS ether by treatment with tBAF as previously described afforded the title compound as an oil, 35 mg, yield 11%. IR (NaCl ν max): 1731 (C=O), 1739, (C=O) 3434 (OH) cm^−1^. ^1^H NMR (400 MHz, CDCl_3_): δ1.99 (s, 3H), 3.73 (s, 6H), 3.75 (s, 3H), 3.81 (s, 3H), 4.23 (d, *J* = 1.98 Hz, 1H), 5.12 (d, *J* = 1.99 Hz, 1H), 6.08 (s, 2H), 6.37–6.57 (m, 3H). ^13^C NMR (100 MHz, CDCl_3_): δ 23.14, 53.93, 55.71, 60.87, 65.03, 61.71, 93.81, 111.21, 113.02, 118.21, 130.98, 131.02, 134.27, 147.12, 147.34, 153.27, 164.45, 181.23. HRMS: found 402.1463 (M^+^+H); C_21_H_24_NO_7_ requires 402.1553.

#### 4.1.17. 3-Ethylidene-4-(3-hydroxy-4-methoxy-phenyl)-1-(3,4,5-trimethoxy-phenyl)-azetidin-2-one (**12**)

To a solution of the silyl ether protected β-lactam **10i** (1 mmol) in DCM (10 mL), stirring at 0 °C under N_2_, was added PPh_3_ (1 mmol) and DEAD (1.2 mmol). Stirring at 0 °C was continued for a further 3 min before the reaction was allowed to warm to room temperature. Diethyl ether (30 mL) was added to the reaction mixture to precipitate the triphenylphosphine oxide side product which was removed by filtration. The filtrate was collected and evaporated to dryness under reduced pressure to afford the product. Separation of the *E*/*Z* isomers was carried out on a Biotage SP1 system using a gradient elution from 2% ethyl acetate in hexanes to 100% ethyl acetate over 20 column volumes, and detection at 280 nm. The product was obtained as a colourless resin, [99], [100], 87 mg, yield 52%. IR (NaCl ν max): 1738 (C=O), 2935 (CH), 3327 (OH) cm^−1^. ^1^H NMR (400 MHz, CDCl_3_): *E isomer*, δ 1.62 (d, *J* = 4.40 Hz, 3H), 3.76 (s, 6H), 3.78 (s, 3H), 3.92 (s, 3H), 5.32 (s, 1H), 5.68, s, 1H), 6.32 (m, 1H), 6.61 (s, 2H), 6.86 (d, *J* = 8.2 Hz, 1H), 7.03 (d, *J* = 1.96 Hz, 1H), 6.99 (m, 1H). *Z isomer*, δ 2.05 (d, *J* = 4.16 Hz, 3H), 3.76 (s, 6H), 3.79 (s, 3H), 3.92 (s, 3H), 5.19 (s, 1H), 5.65–5.66 (m, 2H), 6.63 (s, 2H), 6.85–6.90 (m, 3H). ^13^C NMR (100 MHz, CDCl_3_): *E isomer* δ 13.26, 55.80, 55.88, 60.76, 62.72, 94.47, 110.68, 113.04, 118.87, 123.32, 129.72, 133.94, 134.20, 142.32, 146.04, 146.74, 153.33, 161.19. *Z isomer δ* 14.30, 55.81, 55.82, 60.81, 62.80, 94.57, 110.61, 112.73, 118.48, 126.89, 130.37, 133.93, 134.15, 141.71, 146.11, 146.72, 153.38, 161.79. HRMS: found 408.1411(M^+^+Na); C_21_H_23_NO_6_Na requires 408.1423.

#### 4.1.18. 3-(1,2-Dihydroxyethyl)-4-(3-hydroxy-4-methoxyphenyl)-1-(3,4,5-trimethoxyphenyl) azetidin-2-one (**13**)

To a solution of the silyl ether protected azetidin-2-one (**7o**) (156 mg, 0.312 mmol) in pyridine (0.5 mL) stirring under N_2_ at room temperature was added osmium tetroxide, OsO_4_ (80 mg, 0.312 mmol). The reaction darkened in colour and became hot to the touch upon completion of the addition. The flask was immersed in ice-water for 60 s, then left to stir at room temperature under N_2_ for 22 h. A solution of Na_2_(SO_3_)_2_ (1.343 g, 6.8 mmol) in a 1:4 mixture of pyridine/water (20 mL) was added and the reaction was stirred at room temperature for a further 7 h. The reaction mixture was extracted with warm ethyl acetate (100 mL). The organic layer was collected and washed with 0.1M HCl (100 mL), saturated NaHCO_3_ (100 mL), and water (100 mL). The organic layer was collected and dried over MgSO_4_, filtered and the solvent removed under reduced pressure. The product was purified by passage through a short silica column (5 cm) and eluted with DCM. The tBDMS group was cleaved by treatment with tBAF as described above, to afford the product as a colourless resin, yield 39%, 51 mg. IR (NaCl ν max): 1727 (C=O), 3454 (OH) cm^−1^. ^1^H NMR (400 MHz, CDCl_3_): δ 2.72 (br s, 1H, OH), 3.16 (dd, 0.72H, *J* = 2.42 Hz, 5.55 Hz), 3.19 (dd, 0.29H, *J* = 2.46 Hz, 4.56 Hz), 3.60-3.67 (m, 8H), 3.74 (s, 3H), 3.83 (s, 3H), 4.10 (dd, *J* = 4.05 Hz, 0.26H), 4.19 (dd, *J* = 5.50 Hz, 0.75H), 4.25 (br s, 0.5H), 4.90 (d, *J* = 2.37 Hz, 0.31H), 5.00 (d, *J* = 2.36 Hz, 0.69H), 6.25 (br, s, 0.30H), 6.46 (br, s, 0.70H), 6.53 (s, 2H), 6.77–6.94 (m, 3H). ^13^C NMR (100 MHz, CDCl_3_): δ 55.79, 55.85, 56.80, 56.97, 60.73, 62.18, 64.46, 64.76, 68.54, 69.30, 94.92, 111.06, 111.11, 112.23, 112.43, 117.76, 117.90, 129.92, 130.29, 133.17, 133.33, 146.07, 146.22, 146.85, 146.98, 153.27, 165.76, 165.96. HRMS: found 442.1490 (M^+^+Na); C_21_H_25_NO_8_Na requires 442.1478.

#### 4.1.19. (1-((2-Methoxy-5-(4-oxo-1-(3,4,5-trimethoxyphenyl)-3-vinylazetidin-2-yl)phenyl)-amino)-1-oxopropan-2-yl) carbamaic acid 9H-fluoren-9-ylmethyl ester (**14**)

To a stirred solution of β-lactam **7t** (4.76 mmol) in anhydrous DMF (30 mL) were added DCC (5.7 mmol), Fmoc-protected alanine (5.6 mmol) and HOBt.H_2_O (7.3 mmol) at room temperature. The mixture was stirred for 24 h, then ethyl acetate (50 mL) was added and the reaction mixture was filtered. The DMF was removed by washing with water (5 × 50 mL). The organic solvent was removed under reduced pressure, and the product was isolated by flash column chromatography over silica gel (eluent: dichloromethane: methanol gradient) as a brown oil, yield, 58%, 173 mg. IR (NaCl, film) ν_max_: 3323 (NH), 1723 (C=O), 1640 (C=O), 1598 cm^−1^. H NMR (400 MHz, CDCl_3_): δ 1.52 (br s, 3H), 3.74 (s, 6H), 3.77 (s, 4H), 3.83 (s, 3H), 4.25 (t, *J* = 6.78 Hz, 1H), 4.45–4.47 (m, 3H), 4.76 (br s, 1H), 5.31–5.34 (m, 1H), 5.37–5.41 (m, 1H), 5.98–6.07 (m, 1H), 6.59 (s, 2H), 6.86–7.79 (m, 11H), 8.39–8.44 (br s, 1H), 8.50 (s, 1H). ^13^C NMR (100 MHz, CDCl_3_): δ 20.62, 46.63, 55.47, 55.64, 60.49, 59.96, 60.93, 63.32, 66.83, 94.30, 110.15, 117.70, 119.50, 120.58, 124.52, 126.64, 127.35, 128.39, 128.80, 128.84, 129.31, 130.03, 133.38, 133.91, 137.63, 140.85, 143.22, 153.03, 164.91, 169.98, 170.76. HRMS: Found 700.2632 (M^+^+Na); C_39_H_39_N_3_O_8_Na requires 700.2635.

#### 4.1.20. 2-Amino-N-(2-methoxy-5-(1-(3,4,5-trimethoxyphenyl)-4-oxo-3-vinylazetidin-2-yl)phenyl)propanamide (**15**)

To amino acid amide **14** (1.56 mmol) in methanol (10 mL)/CH_2_Cl_2_ (10 mL) was added 2N NaOH (3.4 mmol) at room temperature and the mixture was stirred for 24 h. Saturated aq. NaHCO_3_ was added and the mixture was extracted with CH_2_Cl_2_ three times. The organic solution was dried and evaporated. The product was dissolved diethyl ether and extracted with 2N HCl (5 × 50 mL). 2N NaOH was added to the HCl mixture solution and the mixture was washed with diethyl ether (5 × 50 mL). The organic solution was dried and the solvent was removed under reduced pressure to afford the product as an off-yellow oil, yield 57%. IR (NaCl) ν_max_: 3307 cm^−1^ (NH_2_), 1741 (C=O), 1679 (C=O) cm^−1^. ^1^H NMR (400 MHz, CDCl_3_): δ 1.61–1.64 (m, 3H), 3.76 (s, 9H), 3.68–3.69 (m, 1H), 3.78 (br s, 1H), 3.79 (s, 3H), 5.40 (br s, 1H), 6.33–6.35 (m, 1H), 6.65 (s, 2H), 6.83–6.90 (m, 2H), 7.08–7.20 (m, 3H). ^13^C NMR (100 MHz, CDCl_3_): δ 21.65, 51.69, 55.87, 55.90, 60.92, 62.95, 65.87, 94.47, 109.81, 119.33, 121.61, 123.70, 127.72, 133.57, 133.69, 134.14, 148.34, 153.45, 161.44, 167.06. HRMS: found 456.2137 (M^+^+H); C_24_H_30_N_3_O_6_ requires 456.2135.

#### 4.1.21. General procedure III: Preparation of dibenzyl phosphates **16a**, **16b**

To a solution of phenol **7s**, **9a** (17 mmol) in acetonitrile (100 mL cooled to 0 °C) was added carbon tetrachloride (85 mmol). The resulting solution was stirred for 10 min prior to adding diisopropylethylamine (35 mmol) and dimethylaminopyridine (1.7 mmol). The dibenzyl phosphite (24.5 mmol) was then added dropwise to the mixture. When the reaction was complete, 0.5M KH_2_PO_4_ (aq) was added and the reaction mixture was allowed to warm to room temperature. An ethyl acetate extract (3 × 50 mL) was washed with saturated sodium chloride (aqueous, 100 mL) followed by water (100 mL) and dried using anhydrous sodium sulfate. The organic solvent was removed under reduced pressure and the product was isolated by flash column chromatography over silica gel (*n*-hexane: ethyl acetate gradient).

##### 2-Methoxy-5-(1-(3,4,5-trimethoxyphenyl)-4-oxoazetidin-2-yl)phenyl dibenzyl phosphate (**16a**)

Preparation as described in the general method above from β-lactam **9a**. Yield: 60%, 507 mg, brown oil. IR (NaCl, film) ν_max_: 2940, 1730 (C=O, β-lactam), 1507, 1300 (P=O), 1235, 1127 cm^−1^. ^1^H NMR (400 MHz, CDCl_3_): δ 2.90 (dd, *J* = 15.08 Hz, 2.52 Hz, 1H), 3.50 (dd, *J* = 15.56 Hz, 5.52 Hz, 1H), 3.76 (s, 6H), 3.76 (s, 3H), 3.81 (s, 3H), 4.84 (dd, *J* = 5.04 Hz, 2.52 Hz, 1H), 5.15–5.18 (m, 4H), 6.53 (s, 2H), 6.93–7.23 (m, 3H), 7.31–7.36 (m, 10H). ^13^C NMR (100 MHz, CDCl_3_): δ 46.44, 53.23, 55.54, 55.61, 60.47, 69.44, 69.48, 69.54, 93.97, 112.77, 119.36, 122.59, 127.43, 127.47, 128.14, 128.16, 130.14, 133.47, 133.89, 135.10, 139.47, 139.54, 150.35, 153.06, 163.87. HRMS: found 642.1838 (M^+^+Na); C_33_H_34_NO_9_PNa requires 642.1869.

##### 2-Methoxy-5-(1-(3,4,5-trimethoxyphenyl)-4-oxo-3-vinylazetidin-2-yl)phenyl dibenzyl phosphate (**16b**)

Preparation as described in the general method above from β-lactam **7s**. Yield: 61%, 502 mg, brown oil. IR (NaCl, film) ν_max_: 2946, 1749 (C=O, β-lactam), 1502, 1300 (P=O), 1240, 1127 cm^−1^. ^1^H NMR (400 MHz, CDCl_3_): δ 3.70 (br s, 1H), 3.71 (s, 6H), 3.76 (s, 3H), 3.82 (s, 3H), 4.65 (d, *J* = 2.00 Hz, 1H), 5.14–5.18 (m, 4H), 5.33–5.40 (m, 2H), 5.97–6.02 (m, 1H), 6.53 (s, 2H), 6.94–7.20 (m, 3H), 7.28–7.35 (m, 10H). ^13^C NMR (100 MHz, CDCl_3_): δ 55.56, 55.61, 60.43, 60.48, 63.29, 69.48, 69.50, 94.19, 112.79, 119.27, 119.62, 122.75, 127.41, 127.49, 128.15, 128.22, 129.29, 129.85, 133.22, 134.00, 135.07, 135.12, 139.57, 150.52, 153.07, 164.69. HRMS: found 668.2017 (M^+^+Na); C_35_H_36_NO_9_PNa requires 668.2025.

#### 4.1.22. Phosphoric acid 2-methoxy-5-[4-oxo-1-(3,4,5-trimethoxyphenyl)azetidin-2-yl]phenyl ester dimethyl ester (**16c**)

A solution of β-lactam phenol **7s** (280 mg, 0.64 mmol), acetonitrile (5 mL) and carbon tetrachloride (0.62 mL, 0.64 mmol) was cooled to −10 °C and stirred under a nitrogen atmosphere for ten minutes. Diisopropyl ethylamine (1.28 mmol) and dimethylaminopyridine (0.06 mmol) were added. After one minute, dimethyl phosphite (0.96 mmol) was added over three minutes. The mixture was stirred for a further 3 h allowing the reaction to come to ambient temperature slowly. The reaction was terminated via the addition of 0.5 M potassium dihydrogen phosphate. The mixture was extracted with ethyl acetate. The organic phases were combined and evaporated to dryness under reduced pressure. The residue was purified by flash chromatography on silica gel to afford the product (155 mg, 52%). IR (KBr) ν_max_: 3437.4, 2960.9, 1752.1, 1603.4, 1509.2, 1466.2, 1281.4, 1239.3, 1185.6, 1130.0, 1052.1, 999.4, 855.8. ^1^H-NMR (400 MHz, CDCl_3_): δ 2.94 (dd, 1H, *J* = 2.8 Hz, 15.3 Hz), 3.52 (dd, 1H, *J* = 5.5 Hz, 15.3 Hz), 3.72 (s, 6H), 3.74 (s, 3H), 3.82–3.87 (m, 9H, OCH_3_), 4.92 (m, 1H), 6.52 (s, 2H), 6.95–6.96 (m, 1H), 7.16–7.18 (m, 1H), 7.29–7.30 (m, 1H). ^13^C NMR (100 MHz, CDCl_3_): δ 46.80, 53.61, 54.91, 54.97, 55.98, 56.05, 60.83, 94.40, 113.29, 119.61, 119.64, 123.21, 130.61, 133.84, 134.23, 139.81, 139.89, 150.72, 150.76, 153.43, 164.37. HRMS: Found 468.1425 (M^+^+H), C_21_H_27_NO_9_P requires 468.1423.

#### 4.1.23. Phosphoric acid diethyl ester 2-methoxy-5-[4-oxo-1-(3,4,5-trimethoxyphenyl)azetidin-2-yl]phenyl ester (**16d**)

Preparation as described above for β-lactam **16c** using diethyl phosphite (0.96 mmol). Yield 250 mg, 79%. IR (KBr) ν_max_: 3487.5, 2988.0, 1756.3, 1603.4, 1586.6, 1507.1, 1451.5, 1292.0, 1238.7, 1127.0, 1035.6, 1000.2, 988.1, 823.1. ^1^H-NMR (400 MHz, CDCl_3_): δ 1.29 (t, 6H), 2.90 (dd, 1H, *J* = 2.5 Hz, 15.3 Hz), 3.49 (dd, 1H, *J* = 5.8 Hz, 15.3 Hz), 3.69 (s, 6H), 3.71 (s, 3H), 3.83 (s, 3H), 4.03 –4.07 (m, 4H), 6.50 (s, 2H), 6.92 (m, 1H), 7.12–7.15 (m, 1H), 7.29–7.30 (m, 1H). ^13^C NMR (100 MHz, CDCl_3_): δ: 15.96, 16.03, 46.84, 53.65, 56.00, 60.84, 63.45, 63.50, 94.38, 113.17, 119.49, 119.51, 123.07, 130.53, 133.88, 134.24, 140.06, 140.13, 150.80, 150.85, 153.45, 164.36. HRMS: Found 496.1734 (M^+^+H); C_23_H_31_NO_9_P requires 496.1736.

#### 4.1.24. 2-Methoxy-5-(1-(3,4,5-trimethoxyphenyl)-4-oxoazetidin-2-yl)phenyl dihydrogen phosphate (**17a**)

Dibenzyl phosphate ester (**16a**) (0.27 mmol) was dissolved in dry dichloromethane (5 mL) under nitrogen at 0 °C. Bromotrimethylsilane (0.59 mmol) was added to the reaction mixture and allowed to stir for 45 min. Sodium thiosulfate solution (10%, 5 mL) was added to the reaction and stirring was continued for 5 min. The aqueous phase was extracted with ethyl acetate (3 × 25 mL). The combined organic phases were concentrated under reduced pressure and the crude product was purified by flash chromatography on silica gel (eluent: *n*-hexane: ethyl acetate, 1:1) to afford the product as a brown solid. Yield: 91%, 289 mg, Mp 207–209 °C. Purity (HPLC): 100.0%. IR (KBr) ν_max_: 3497 (OH), 1730 (C=O, β-lactam), 1303 (P=O), 1237, 1128 cm^−1^. ^1^H NMR (400 MHz, DMSO-*d_6_*): δ 2.91 (dd, *J* = 15.04 Hz, 2.52 Hz, 1H), 3.51 (dd, *J* = 15.64 Hz, 5.52 Hz, 1H), 3.58 (s, 3H), 3.64 (s, 6H), 3.74 (s, 3H), 5.08 (br s, 1H), 6.52 (s, 2H), 7.03–7.48 (m, 3H). ^13^C NMR (100 MHz, DMSO-*d_6_*): δ 45.88, 52.92, 55.64, 55.72, 60.06, 94.30, 113.02, 118.96, 121.91, 130.08, 130.71, 133.53, 133.68, 150.19, 153.09, 164.47. HRMS: found 438.0947 (M-H)^-^; C_19_H_21_NO_9_P requires 438.0954

#### 4.1.25. 2-Methoxy-5-(1-(3,4,5-trimethoxyphenyl)-4-oxo-3-vinylazetidin-2-yl)phenyl dihydrogen phosphate (**17b**)

Following the preparation described above for compound **16a**, using dibenzyl phosphate ester (**16b**) (0.27 mmol) and bromotrimethylsilane (0.59 mmol). Purification by flash chromatography on silica gel (eluent: *n*-hexane: ethyl acetate, 1:1) afforded the product as a yellow oil, yield 63%. IR (NaCl) ν_max_: 3483 (OH), 1749 (C=O), 1307 (P=O) cm^−1^. ^1^H NMR (400 MHz, CDCl_3_): δ 3.70 (s, 6H), 3.76 (s, 3H), 3.79 (s, 3H), 4.68–4.70 (m, 1H), 5.14 (br s, 1H), 5.28–5.38 (m, 2H), 5.94–6.00 (m, 1H), 6.51 (s, 2H), 6.91–7.35 (m, 3H). ^13^C NMR (100 MHz, CDCl_3_): δ 55.49, 55.58, 60.43, 60.50, 63.07, 94.31, 112.69, 119.18, 119.55, 122.70, 129.81, 133.08, 133.99, 134.67, 135.23, 150.52, 153.02, 165.14. HRMS: found 488.1106 (M^+^+Na); C_21_H_24_NO_9_PNa requires 488.1086.

#### 4.1.26. 5-(3-Ethyl-1-(3,4,5-trimethoxyphenyl)-4-oxoazetidin-2-yl)-2-methoxyphenyl dihydrogen phosphate (**17c**)

The dibenzylphosphate ester protected compound **16b** (2 mmol) was dissolved in ethanol: ethyl acetate (50 mL; 1:1 mixture) and hydrogenated over 1.2 g of 10% palladium on carbon until complete on TLC, typically less than 3 h. The catalyst was filtered, the solvent was removed under reduced pressure and the product was isolated by flash column chromatography over silica gel (eluent: *n*-hexane: ethyl acetate gradient) to afford the product as a brown oil, 140 mg, 98%. Purity (HPLC): 100%. IR (NaCl, film) ν_max_: 3483 (OH), 1742 (C=O), 1272 (P=O), 1236, 1126 cm^−1^. ^1^H NMR (400 MHz, CDCl_3_): δ 0.97 (br s, 3H), 1.76–1.83 (m, 2H), 3.09 (s, 1H), 3.66 (s, 6H), 3.71 (s, 6H), 4.56 (br s, 1H), 6.51 (s, 2H), 6.82–7.38 (m, 3H). ^13^C NMR (100 MHz, CDCl_3_): δ 13.74, 20.62, 55.47, 55.55, 59.67, 59.99, 60.39, 94.33, 112.48, 118.69, 122.27, 130.12, 133.16, 133.86, 135.64, 150.06, 152.97, 168.22. HRMS: found 490.1239 (M^+^+Na); C_21_H_26_NO_9_PNa requires 490.1243.

### 4.2. Biochemical Evaluation

All biochemical assays were performed in triplicate on at least three independent occasions for the determination of mean values reported.

#### 4.2.1. Cell Culture

The human breast carcinoma cell line MCF-7, was purchased from the European Collection of Animal Cell Cultures (ECACC) and was cultured in Eagle’s minimum essential medium with 10% fetal bovine serum, 2 mM L-glutamine and 100 μg/mL penicillin/streptomycin. The medium was supplemented with 1% non-essential amino acids. The human breast carcinoma cell line MDA-MB-231 was purchased from the European Collection of Animal Cell Cultures (ECACC). MDA-MB-231 cells were maintained in Dulbecco’s modified Eagle’s medium (DMEM) supplemeted with 10% (*v*/*v*) fetal bovine serum, 2 mM L-glutamine and 100 μg/mL penicillin/streptomycin (complete medium). All media contained 100 U/mL penicillin and 100 μg/mL streptomycin. Triple negative breast cancer Hs578T cells and its invasive variant Hs578Ts(i)_8_ were obtained as a kind gift from Dr. Susan McDonnell, School of Chemical and Bioprocess Engineering, University College Dublin and were cultured in Dulbecco’s Modified Eagle’s Media (DMEM) with GlutaMAX^TM^-I, with the same supplement as for MDA-MB-231 cells in the absence of non-essential amino acids. HEK-293T normal epithelial embryonic kidney cells were cultured in Dulbecco’s Modified Eagle’s Medium (DMEM) with GlutaMAX^TM^-I in the absence of non-essential amino acids. K562 and HL-60 cells were originally obtained from the European Collection of Cell Cultures (Salisbury, UK).The K562 cells were derived from a patient in the blast crisis stage of CML HL-60 cells were derive from a patient with acute myeloid leukaemia. Cells were cultured in RPMI-1640 Glutamax medium supplemented with 10% FCS media, and 100 μg/mL penicillin/streptomycin. Cells were maintained at 37 °C in 5% CO_2_ in a humidified incubator. All cells were sub-cultured three times/week (adherent cells by trypsinisation).

#### 4.2.2. Cell Viability Assay

Cells were seeded at a density of 5 × 10^3^ cells/well (MCF-7), in triplicate in 96-well plates. After 24 h, cells were then treated with either medium alone, vehicle [1% ethanol (*v*/*v*)] or with serial dilutions of CA-4 or β-lactam analogue. Cell viability for MCF-7 and MDA-MB-231 was analysed using the Alamar Blue assay (Invitrogen Corp, Thermo Fisher Scientific, 168 Third Avenue, Waltham, MA, USA) according to the manufacturer’s instructions. After 72 h, Alamar Blue [10% (*v*/*v*)] was added to each well and plates were incubated for 3–5 h at 37 °C in the dark. Fluorescence was read using a 96-well fluorimeter with excitation at 530 nm and emission at 590 nm. Results were expressed as percentage viability relative to vehicle control (100%). Dose response curves were plotted and IC_50_ values (concentration of drug resulting in 50% reduction in cell survival) were obtained using the commercial software package Prism (GraphPad Software, Inc., 2365 Northside, Suite 560, San Diego, CA, USA). Experiments were performed in triplicate on at least three separate occasions.

#### 4.2.3. Lactate Dehydrogenase Assay for Cytotoxicity

Cytotoxicity was determined using the CytoTox 96 non-radioactive cytotoxicity assay (Promega Corporation; 2800 Woods Hollow Road, Madison, WI, USA) [101] following the manufacturer’s protocol. Briefly, MCF-7 cells were seeded in 96-well plates, incubated for 24 hr and then treated with test compounds (**7d**, **7h**, **7i**, **7q**, **7r**, **7u**, **7t**, **10d**, **10f**, **12**) as described in the cell viability assay above. After 72 h, 20 μL of ‘lysis solution (10X)’ was added to control wells and the plate was incubated for a further 1 hr to ensure 100% death. 50 μL of supernatant was carefully removed from each well and transferred to a new 96-well plate. 50 μL of reconstituted ‘substrate mix’ was added and the plate was placed in the dark at room temperature for 30 min. After this period, 50 μL of ‘stop solution’ was added to each well and the absorbance was read at a wavelength of 490 nm using a Dynatech MR5000 plate reader. The percentage cell death at 10 μM was calculated.

#### 4.2.4. Cytotoxicity Assay

As previously reported [45,74,75] mammary glands from 14–18 day pregnant CD-1 mice were used as source and primary mammary epithelial cell cultures were prepared from these. The isolated mammary epithelial cells were seeded at two concentrations (25,000 cells/mL and 50,000 cells/mL). Initially a third concentration of 100,000 cells/mL was also used, but this proved to be too high to give meaningful results. After 24 h, the cells were treated with 2 µL volumes of test compound **7s** which had been pre-prepared as stock solutions in ethanol to furnish the concentration range of study, 1 nM–100 µM, and re-incubated for a further 72 h. Control wells contained the equivalent volume of the vehicle ethanol (1% *v*/*v*). The cytotoxicity was assessed using alamar blue dye.

#### 4.2.5. Cell Cycle Analysis

MCF-7 cells (adherent and detached) were treated with the appropriate concentration of compound **7s** and incubated for the designated time. Cells were collected, trypsinised and centrifuged at 800× *g* for 15 min. Cells were washed twice with ice-cold PBS and fixed in ice-cold 70% ethanol overnight at −20 °C. Fixed cells were centrifuged at 800× *g* for 15 min and stained with 50 μg/mL of PI, containing 50 μg/mL of DNase-free RNase A, at 37 °C for 30 min. The DNA content of cells (10,000 cells/experimental group) was analysed by flow cytometer at 488 nm using a FACSCalibur flow cytometer (BD Biosciences, 2350 Qume Dr, San Jose, CA, USA) all data were recorded and analysed using the CellQuest^TM^ software, (BD Biosciences, 2350 Qume Dr, San Jose, CA, USA)

#### 4.2.6. Annexin V/PI Apoptotic Assay

Apoptotic cell death was detected by flow cytometry using Annexin V and propidium iodide (PI). MCF-7 Cells were seeded in 6 well plated at density of 1 × 10^5^ cells/mL and treated with either vehicle (0.1% (*v*/*v*) EtOH), CA-4 or β-lactam compound **7s** at different concentrations for selected time. Cells were then harvested and prepared for flow cytometric analysis. Cells were washed in 1X binding buffer (20X binding buffer: 0.1M HEPES, pH 7.4; 1.4 M NaCl; 25 mM CaCl_2_ diluted in dH_2_O) and incubated in the dark for 30 min on ice in Annexin V-containing binding buffer [1:100]. Cells were then washed once in binding buffer and then re-suspended in PI-containing binding buffer [1:1000]. Samples were analysed immediately using the BD accuri flow cytometer (BD Biosciences, 2350 Qume Dr, San Jose, CA, USA) and prism software for analysis the data (GraphPad Software, Inc., 2365 Northside Dr., Suite 560, San Diego, CA, USA). Four populations are produced during the assay Annexin V and PI negative (Q4, healthy cells), Annexin V positive and PI negative (Q3, early apoptosis), Annexin V and PI positive (Q2, late apoptosis) and Annexin V negative and PI positive (Q1, necrosis). Paclitaxel was used as a positive control for cell death

#### 4.2.7. Tubulin Polymerization Assay

The assembly of purified bovine tubulin was monitored using a kit, BK006, purchased from Cytoskeleton Inc., 1830 S Acoma St, Denver, CO, 80223, USA. [78]. The assay was carried out in accordance with the manufacturer’s instructions in the tubulin polymerisation assay kit manual using the standard assay conditions. The values reported represent the average values from two independent assays. Purified (>99%) bovine brain tubulin (3 mg/mL) in a buffer consisting of 80 mM PIPES (pH 6.9), 0.5 mM EGTA, 2 mM MgCl_2_, 1 mM GTP and 10% glycerol was incubated at 37 °C in the presence of either vehicle (2% (*v*/*v*) ddH_2_O) or compounds **7h**, **7i**, **7s**, **7t** (initially 10 µM in EtOH); CA-4 and Paclitaxel were used as controls. Light is scattered proportionally to the concentration of polymerised microtubules in the assay. Therefore, tubulin assembly was monitored turbidimetrically at 340 nm at 37 °C in a Spectramax 340 PC spectrophotometer (Molecular Devices, 3860 N 1st St, San Jose, CA, USA). The absorbance was measured at 30 s intervals for 60 min.

#### 4.2.8. Colchicine-Binding Site Assay

MCF-7 cells were seeded at a density of 5 × 10^4^ cells/well in 6-well plates and incubated overnight. Cells were treated with vehicle control [ethanol (0.1% *v*/*v*)] or compound **7s** (10 μM) for 2 h. After this time, selected wells were treated with *N*,*N*′-ethylene-bis(iodoacetamide)(EBI) (100 μM) (Santa Cruz Biotechnology Inc. 10410 Finnell Street, Dallas, Texas, USA) for 1.5 h. Following treatment, cells were twice washed with ice-cold PBS and lysed by addition of Laemmli buffer. Samples were separated by SDS-PAGE, trasnsferred to polyvinylidene difluoride membranes and probed with β-tubulin antibodies (Sigma-Aldrich, 2033 Westport Center Dr, St. Louis, MO, USA) [85].

#### 4.2.9. Immunofluorescence Microscopy

Confocal microscopy was used to study the effects of drug treatment on MCF-7 cytoskeleton. For immunofluorescence, MCF-7 cells were seeded at 1 × 10^5^ cells/mL on eight chamber glass slides (BD Biosciences, 2350 Qume Dr, San Jose, CA, USA). Cells were either untreated or treated with vehicle [1% ethanol (*v*/*v*)], paclitaxel (1 μM), combretastatin A-4 (100 nM) or compound **7s** (100 nM) for 16 h. Following treatment cells were gently washed in PBS, fixed for 20 min with 4% paraformaldehyde in PBS and permeabilised in 0.5% Triton X-100. Following washes in PBS containing 0.1% Tween (PBST), cells were blocked in 5% bovine serum albumin diluted in PBST. Cells were then incubated with mouse monoclonal anti-α-tubulin−FITC antibody (clone DM1A) (Sigma-Aldrich, 2033 Westport Center Dr, St. Louis, MO, USA) (1:100) for 2 h at room temperature. Following washes in PBST, cells were incubated with Alexa Fluor 488 dye (1:450) for 1 h at room temperature. Following washes in PBST, the cells were mounted in Ultra Cruz Mounting Media (Santa Cruz Biotechnology Inc., 10410 Finnell Street, Dallas, TX, USA) containing 4,6-diamino-2-phenolindol dihydrochloride (DAPI). Images were captured by Leica SP8 confocal microscopy with Leica application suite X software (Leica Microsystems CMS GmbH Am Friedensplatz 3 D-68165 Mannheim, Germany). All images in each experiment were collected on the same day using identical parameters. Experiments were performed on three independent occasions.

### 4.3. Stability Study of Compounds 7s, 17b and 17c

Analytical high-performance liquid chromatography (HPLC) stability studies were performed using a Symmetry^®^ column (C_18_, 5 µm, 4.6 × 150 mm), a Waters 2487 Dual Wavelength Absorbance detector, a Waters 1525 binary HPLC pump and a Waters 717 plus Autosampler (Waters Corporation, 34 Maple St, Milford, MA, USA). Samples were detected at wavelength of 254 nm. All samples were analysed using acetonitrile (80%) and water (20%) as the mobile phase over 10 min and a flow rate of 1 mL/min. Stock solutions were prepared by dissolving 5 mg of compound **7s**, **17b** and **17c** in 10 mL of mobile phase. Phosphate buffers at the desired pH values (4, 7.4 and 9) were prepared in accordance with the British Pharmacopoeia monograph 2015. 30 µL of stock solution was diluted with 1 mL of appropriate buffer, shaken and injected immediately. Samples were withdrawn and analysed at time intervals of *t* = 0 min, 5 min, 30 min, 60 min, 90 min, 120 min, 24 h and 48 h. **Plasma stability studies**: 360 µL stock solution of compounds **7s**, **17b** and **17c** were transferred to buffered plasma (plasma: buffer = 1:9, 4 mL in total) at 37 °C in screw cap container. Immediately a 250 µL aliquot was withdrawn and added to the Eppendorf tube containing 500 µL ZnSO_4_.7H_2_O solution (2% w/v ZnSO_4_ solution in acetonitrile: water, 1:1). The samples were then centrifuged at 10,000 rpm for 3 min and filtered through a 0.2-micron filter and injected according to the HPLC conditions listed above. Further samples were taken in the same manner every 1 h thereafter up to 6 h. A final sample was taken after 24 h. **Whole blood stability studies**. A 360 µL aliquot of stock solution of compounds **17b** and **17c** in acetonitrile was added in whole blood (4 mL, treated with 2% sodium citrate) at 37 °C and 300 µL aliquots were withdrawn at approptiate intervals. Samples were transferred to 1.5 mL Eppenddorf tubes containing 1 mL of ZnSO_4_. 7H_2_O solution (2% w/v ZnSO_4_ solution in acetonitrile: water, 1:1), vortexed and then centrifuged for 5 min at 14,000 rpm. The sample filtered through a 0.2-micron filter and injected according to the HPLC conditions listed above.

### 4.4. X-Ray Crystallography

Data for samples **6k**, **7h** and **8i**, **8k** were collected on a Bruker APEX DUO and Bruker D8 Quest ECO respectively using Cu Kα (λ = 1.54178 Å; **7h**) and Mo Kα radiation (λ = 0.71073 Å; **6k**, **8i**, **8k**), (Bruker, 40 Manning Road, Billerica, MA, USA) Each sample was mounted on a Mitegen cryoloop and data collected at 100(2) K (Oxford Cobra and Cryostream cryosystems, Oxford Cryosystems, 3 Blenheim Office Park, Long Hanborough, Oxford OX29 8LN, UK). Bruker APEX [102] software was used to collect and reduce data and determine the space group. The structures were solved using direct methods (XS) [103] or intrinsic phasing (XT) [104] and refined with least squares minimization (XL) [105] in Olex2 [106]. Absorption corrections were applied using SADABS 2014 [107]. Data for sample **8h** were collected on a Rigaku Saturn 724 at 150(2) K (X-Stream), (Rigaku, Tokyo, Tōkyō Prefecture, JP 151-0051) and CrystalClear [108,109] was used for cell refinement and data reduction and absorption corrections. Bruker APEX software as well as XT, XL were used to determine the space group, solve and refine the structure in Olex2 [106]. Crystal data, details of data collections and refinement are given in Table 1. CCDC 1820354-1820359 contains the supplementary crystallographic data for this paper. These data can be obtained free of charge from The Cambridge Crystallographic Data Centre (www.ccdc.cam.ac.uk/data_request/cif).

All non-hydrogen atoms were refined anisotropically. Hydrogen atoms were assigned to calculated positions using a riding model with appropriately fixed isotropic thermal parameters. Some structures have multiple independent molecules in the asymmetric unit: **7h** has 4 independent molecules with chirality C3: R, C4: S, C30: R, C31: S, C57: S, C58: R, C84: S and C85: R; **8i** has 2 independent molecules with chirality: C3: R, C4: S, C30: R and C3: S.

### 4.5. Computational Procedure: Molecular Docking Study

For ligand preparation, all compounds were built using ChemBioDraw 13.0, (PerkinElmer Inc., 940 Winter Street, Waltham, MA 02451, USA) saved as mol files and opened in MOE (Molecular Operating Environment (MOE) Version 2015.10, Chemical Computing Group Inc., 1010 Sherbrooke St W, Montreal, QC, Canada). For the receptor preparation, the PDB entry1SA0 was downloaded from the Protein Data Bank PDB [5]. A UniProt Align analysis confirmed a 100% sequence identity between human and bovine beta tubulin. All waters were retained in both isoforms. Addition and optimisation of hydrogen positions for these waters was carried out using MOE 2015.10 ensuring all other atom positions remained fixed [110]. For both enantiomers of each compound, MMFF94x partial charges were calculated and each was minimised to a gradient of 0.001 kcal/mol/Å. Default parameters were used for docking except that 500 poses were sampled for each enantiomer and the top 50 docked poses were retained for subsequent analysis. The crystal structure was prepared using QuickPrep (minimised to a gradient of 0.001 kcal/mol/Å), Protonate 3D, Residue pKa and Partial Charges protocols in MOE 2016 with the MMFF94x force field.

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
