# Peer review of "3-Vinylazetidin-2-Ones: Synthesis, Antiproliferative and Tubulin Destabilizing Activity in MCF-7 and MDA-MB-231 Breast Cancer Cells"

_pharmaceuticals, 2019, doi:10.3390/ph12020056_

Round 1

Reviewer 1 Report

Very interesting manuscript about synthesis, characterisation and pharmacolocial evaluation of a variety of 2-azetidinones.

I recommend acceptance in the present form. The only critisism this reviewer has, is he length of the manuscript. Data is sufficient to split the manucript and publish two papers. One about synthesis and characterisation and one about pharmacology.

Author Response

I am happy to proceed with the manuscript in the present form as submitted. 

Reviewer 2 Report

The manuscript of Wang et al report the synthesisi and in vitro biological evaluation of new beta-lacttam derivatives of combretastatin A-4. The manuscript is well written end the result support the conclusion. I have only few and minor comments

1) Since the same research group has recently published a series of other beta-lactam derivatives of CA-4 (ref 46), which show excellent in vitro antiproliferative activity comparable with 7s, it would be useful for the reader to have a comment on the comparison between these compounds and those presented in current manuscript.

2) Figure 6. In adition to representative flow.cytometry plots it would be nice to present a bar-histogram in which these results are collectively showed and statistically analysed.

3) Concerning the induction of apoptosis by 7s, this compound is endowed in two cel lines with a IC50 value of about 10 nM, again from figure 6 it appear that 7s induce only 17.2% and 35.5% of cell death at 10 nM and 100 nM respectively. How can this apparent discrepancy be explained? Have the authors checked whether these compounds, as for other antimicrotubule agents, can induce other forms of cell death and / or autophagy?

4) Considere the very low concentration of many of the new compounds for the sake of clarity the IC50 values should be converted in nM.

Reviewer 3 Report

This study shows the discovery of 3-vinyl-β-lactam 7s and indicates this compound has a promising candidate for development as antiproiferative microtubule-disrupting agents. Therefore, the topic is timely. My biggest concerns about this study are there is no data showing the mechanism of action of compound 7s and lack of a strong rationale for study the drug efficacy in breast cancer cells. Other comments are as follows.

1.      MCF10A (normal human breast epithelial cells), but not HEK293, is the control cell line for breast cancer cells. Authors should compare viability between MCF10A and breast cancer cells after drug treatment.

2.      Since there is no in vivo data, colony formation assays and soft agar assays should better approaches for determination of the long-term effect of drugs on cancer cell growth. Otherwise, only presenting short-term effects of drugs cannot provide compelling evidence that they are promising anticancer candidates.

3.      In figures 6 and 9, quantitative data of apoptotic rate should be included with the representative results.

4. Some references, especial those published in 15 years ago, are not really necessary, which should be removed.   

Round 2

Reviewer 3 Report

Although authors haven’t done some experiment as suggested, they expanded the discussion with these comments which improves the impact of this study. I am happy with this reversion.